# Bulk cell density and Wnt/TGFbeta signalling regulate mesendodermal patterning of human pluripotent stem cells

Henning Kempf[1,2], Ruth Olmer[1,2,3], Alexandra Haase[1,2], Annika Franke[1,2], Emiliano Bolesani[1,2], Kristin Schwanke[1,2], Diana Robles-Diaz[1,2], Michelle Coffee[1,2], Gudrun Göhring[2,4], Gerald Dräger[2,5,6], Oliver Pötz[7], Thomas Joos[7], Erik Martinez-Hackert[8], Axel Haverich[1,2], Falk F.R. Buettner[2,9], Ulrich Martin[1,2,3] & Robert Zweigerdt[1,2]

*In vitro* differentiation of human pluripotent stem cells (hPSCs) recapitulates early aspects of human embryogenesis, but the underlying processes are poorly understood and controlled. Here we show that modulating the bulk cell density (BCD: cell number per culture volume) deterministically alters anteroposterior patterning of primitive streak (PS)-like priming. The BCD in conjunction with the chemical WNT pathway activator CHIR99021 results in distinct paracrine microenvironments codifying hPSCs towards definitive endoderm, precardiac or presomitic mesoderm within the first 24 h of differentiation, respectively. Global gene expression and secretome analysis reveals that TGFß superfamily members, antagonist of Nodal signalling LEFTY1 and CER1, are paracrine determinants restricting PS progression. These data result in a tangible model disclosing how hPSC-released factors deflect CHIR99021-induced lineage commitment over time. By demonstrating a decisive, functional role of the BCD, we show its utility as a method to control lineage-specific differentiation. Furthermore, these findings have profound consequences for inter-experimental comparability, reproducibility, bioprocess optimization and scale-up.

[1] Leibniz Research Laboratories for Biotechnology and Artificial Organs (LEBAO), Department of Cardiothoracic, Transplantation and Vascular Surgery (HTTG), Hannover Medical School, 30625 Hannover, Germany. [2] REBIRTH-Cluster of Excellence, Hannover Medical School, 30625 Hannover, Germany. [3] Biomedical Research in Endstage and Obstructive Lung Disease Hannover (BREATH), German Center for Lung Research (DZL), 30625 Hannover, Germany. [4] Institute of Human Genetics, Hannover Medical School, 30625 Hannover, Germany. [5] Institute of Organic Chemistry, Leibniz University Hannover, 30167 Hannover, Germany. [6] Center of Biomolecular Drug Research (BMWZ), Leibniz University Hannover, 30167 Hannover, Germany. [7] Natural and Medical Sciences Institute at the University of Tuebingen (NMI), 72770 Reutlingen, Germany. [8] Department of Biochemistry and Molecular Biology, Michigan State University, 48824-1319 Michigan, USA. [9] Institute for Cellular Chemistry, Hannover Medical School, 30625 Hannover, Germany. Correspondence and requests for materials should be addressed to R.Z. (email: Zweigerdt.Robert@mh-hannover.de).

Human pluripotent stem cells (hPSCs), including embryonic (hESCs) and induced pluripotent stem cells, provide an attractive model to study early aspects of human embryogenesis *in vitro*, which are not assessable *in vivo*. Moreover, due to their growth and differentiation potential, hPSCs constitute a well-characterized, generally unlimited cell source for the mass generation of lineage- and possibly patient-specific progenies. This opens new avenues for regenerative cell therapies and superior drug discovery approaches. hPSC-based modelling of early human development and the large-scale production of lineage-specific progenies will both depend on well-defined, highly reproducible culture and differentiation conditions[1].

We and others recently showed that hPSCs can be expanded as cell-only aggregates in suspension culture[2–4], thereby providing a flexible strategy for process optimization and scale-up. It enables extensive parameter permutation in multiwell dishes, simple expansion in Erlenmeyer flasks and ultimately transition to more sophisticated, stirred tank bioreactors[5–7].

Initiating differentiation, per definition, turns PSC aggregates into embryoid bodies (EBs) wherein a wide variety of cell types can form. It is generally thought that tissue differentiation in EBs occurs in a disorganized fashion[8]. However, work on mouse EBs showed that the supplementation of external growth factors, such as the WNT pathway agonist Wnt3a, mediates the execution of self-reinforcing, gastrulation-like processes in EBs displaying a high degree of self-organization[9].

During gastrulation, cells are allocated into the three germ layers in an ordered spatiotemporal sequence. Epiblast cells located at the interior of the embryo migrate to form the definitive endoderm on the outside of the embryo proper and the mesoderm between the endoderm and the epiblast. It is well established that the spatial patterning during gastrulation in mouse embryos is under the control of the Activin-Nodal, BMP and WNT pathways, which can also be manipulated to direct mouse and hESCs into derivatives of all three germ layers[10–13].

However, controlling the complex pathways' interplay during PSC differentiation is challenging. It depends on the concentration of respective growth factor combinations as well as their temporal supplementation patterns, and is often accompanied by the heterogeneity of differentiation outcomes lacking inter-experimental reproducibility[14].

Using cardiomyogenic differentiation as central reference point downstream of primitive streak (PS)-like priming, we have focused on modulating the bulk cell density (BCD) defined as the number of hPS cells per medium volume in a respective culture vessel. From a practical perspective, changes of the BCD in conjunction with the GSK3ß inhibitor CHIR99021 (CHIR)[15] can be used as a simple method to direct PS-like patterning into definitive endoderm, precardiac mesoderm (PCM) or presomitic mesoderm (PSM).

We show that BCD modulation only, within the first 24 h, results in distinct gene and protein expression patterns equivalent to specific cell fates along a PS-like anteroposterior axis. Secretome analysis reveals that the BCD effect is mediated via distinct, time-dependent medium conditioning. Functional analysis of secreted candidate factors unravelled that transforming growth factor beta (TGFβ) family members, Nodal signalling antagonists LEFTY1 (left-right determination factor 1) and CERBERUS (CER1) that are either expressed readily at the pluripotent hPSC state or differentially upregulated in response to CHIR, respectively, play decisive roles in PS-like determination of hPSC, reminiscent of their function in mouse embryogenesis[16]. By putting into perspective the impact of the BCD, the CHIR concentration and the incubation time, our data highlight the necessity to closely monitor the BCD, particularly at the earliest stages of differentiation, to ensure desired lineage specification and process reproducibility.

## Results

### BCD and CHIR determine differentiation outcome within 24 h.

Single-cell inoculation resulted in highly reproducible aggregate formation with $\sim 1 \times 10^6$ cells per well in 12-well format (Supplementary Fig. 1a,b) maintaining pluripotency marker expression (Supplementary Fig. 1c), as shown before[2,6].

Differentiation was induced by chemical WNT pathway modulation[17,18] with equal cell numbers per well at initiation. Dissecting the protocol into an early stage (day 0–day 1; 24 h CHIR supplementation) and a late stage (day 1–day 10), two factors, the CHIR concentration and the culture medium volume (defining the BCD), were modified and cardiomyogenesis was monitored at day 10 by NKX2.5-GFP expression[19]. Transgene expression was highly dependent on the CHIR concentration during the first 24 h, as expected[17]. BCD changes at the late stage had no apparent impact (Supplementary Fig. 1d), whereas modulations during the first 24 h had profound consequences. In 1 ml medium per well (BCD: $1 \times 10^6$ cells per ml) higher CHIR concentrations were concomitant with increased green fluorescent protein (GFP) expression (CHIR optimum at 15 µM; Fig. 1b and Supplementary Fig. 1e). Vice versa, in 3 ml medium per well (BCD: $0.33 \times 10^6$ cells per ml), GPF was highest at 7.5 µM CHIR with decreasing levels along increasing CHIR concentrations. In 2 ml per well (BCD: $0.5 \times 10^6$ cells per ml), GFP expression peaked at intermediate CHIR concentrations of 10–12.5 µM. GFP levels correlated with overall cell counts on day 10 (Supplementary Fig. 1f) as noted before[17].

For detailed investigations, four cornerstone conditions were defined (Fig. 1b): two representing high NKX2.5-GFP (efficient cardiomyogenesis): 7.5 µM CHIR in 3 ml (7.5/3; tagged in light green) and 15 µM CHIR in 1 ml (15/1; dark green); and two representing low NKX2.5-GFP: 7.5 µM in 1 ml (7.5/1; blue) and 15 µM in 3 ml (15/3; red). Figure 1c depicts GFP assessment of 11 independent repeats for every condition, revealing the grade of experimental variability; the average amount of GFP$^+$ was 59.3 ± 4.4% (7.5/3), 45.8 ± 5.9% (15/1), 6.7 ± 2.6% (7.5/1) and 20.0 ± 7.3% (15/3), respectively.

Flow cytometry specific to the cardiomyocyte markers cardiac myosin heavy chain, cardiac troponin T and sarcomeric actinin confirmed NKX2.5-GFP results, showing $\sim 67$–74% positivity for 7.5/3 and 15/1, and < 15% positive cells for 7.5/1 and 15/3 conditions (Fig. 1d). Conversely, relative high contents of CD90- and vimentin-expressing cells (31.0–47.4%) were found at 7.5/1 and 15/3 (Supplementary Fig. 1g), suggesting a fibroblast-like phenotype of non-cardiomyocytes.

### Effect of BCD is platform-independent.

Instead of modulating the medium volume at constant cell counts per well (illustrated in Fig. 1e(A)) the cell count per well was modified at equal medium volume (Fig. 1e(B)) and agitation was applied during CHIR exposure (Fig. 1e(B)—agitated) to verify that the 'volume effect' described above is indeed tantamount to the BCD rather than the formation of volume-induced gradients or the like. At 7.5/3 and 15/1 conditions, the highest GFP levels were observed by all experimental variations, strongly supporting that BCD-directed differentiation is platform-independent. Furthermore, differentiation in two-dimensional (2D) monolayer showed an equivalent BCD-dependent GFP pattern reflecting the three-dimensional (3D) aggregate approach (Fig. 1f). Together, this indicates the key impact of BCD modulation during the first 24 h of differentiation irrespective of the culture platform, prompting investigations into the early stages of differentiation.

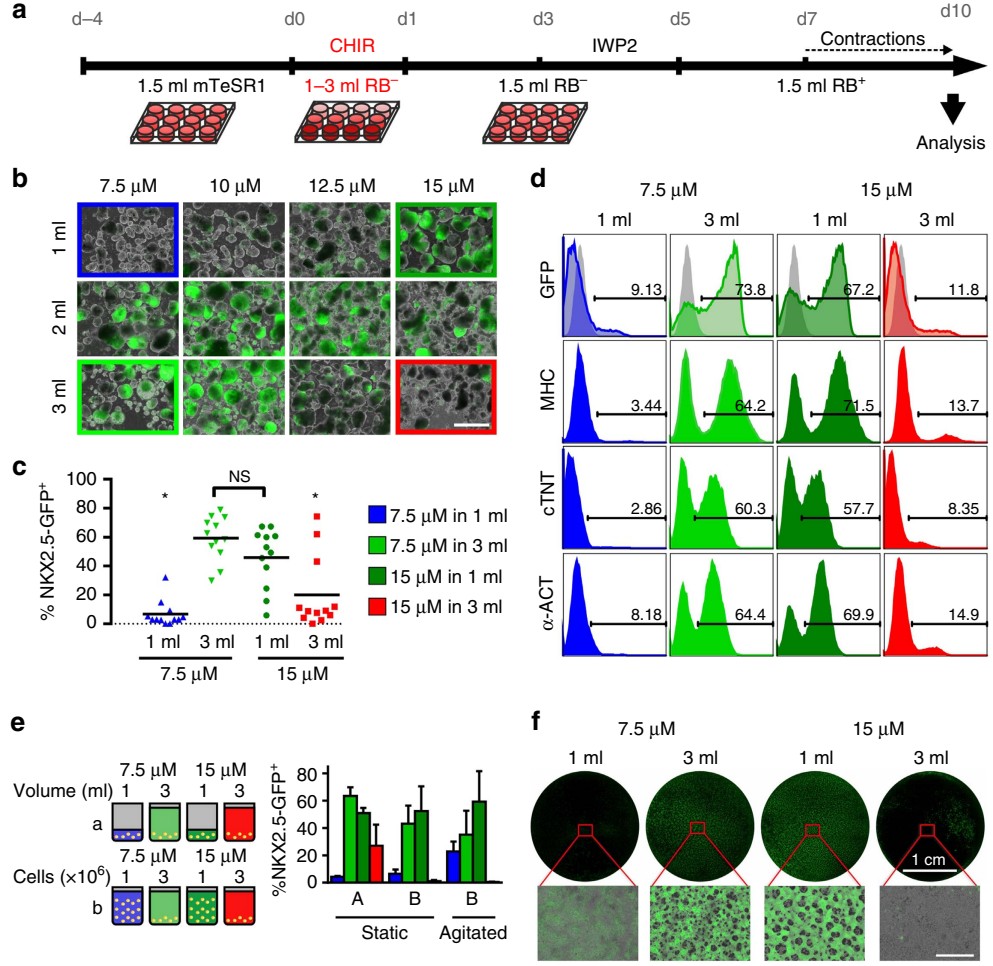

**Figure 1 | The CHIR-to-volume ratio during the first 24 h dictates differentiation outcome.** (**a**) Schematic of expansion and subsequent differentiation. The medium volume of 1–3 ml and consequently the BCD is only modulated between day 0–1. (**b**) Representative images of the differentiation on day 10 using the NKX2.5-GFP cell line. The cornerstone conditions are marked by a coloured rectangle. Scale bar, 1 mm. (**c**) Analysis of NKX2.5-GFP $^+$ on day 10 of the four cornerstone conditions, $n = 11$ independent experiments. The mean is shown as black line. *$P < 0.01$, evaluated by one-way analysis of variance with *post hoc* Bonferroni analysis. (**d**) Representative histograms for NKX2.5-GFP $^+$ and staining against structural cardiac markers on day 10 obtained by flow cytometry. (**e**) Schematic of experiment (left) and corresponding NKX2.5-GFP $^+$ on day 10 (right) in standard conditions using identical cell numbers in different volumes (A) and in identical volume with adapted cell numbers for static as well as agitated conditions (B). Bars represent mean ± s.e.m. of $n = 3$ independent experiments. (**f**) Representative images of NKX2.5-GFP expression on day 10 of differentiation performed in 2D monolayer cultures. Scale bar, 1 mm. See also Supplementary Fig. 1. NS, not significant.

**Cornerstone-specific mesendodermal patterns on day 3.** Epithelial-to-mesenchymal transition of hPSCs and induction of mesoderm commitment can be monitored by the reciprocal epithelial cell adhesion molecule (EpCAM; CD326) to neural cell adhesion molecule (NCAM; CD56) pattern[20,21]. The mesodermal marker NCAM was upregulated 2–3 days after addition of CHIR (Supplementary Fig. 2a), which was accompanied by down-regulation of pluripotency-associated EpCAM (Fig. 2a). On day 3 the lowest NCAM $^+$ levels of 11.3 ± 8.5% were observed at 7.5/1, highest of 79.8 ± 4.3% at 15/3 and intermediate levels of 45.1 ± 8.3% and 51.4 ± 8.9% at 7.5/3 and 15/1, respectively (Fig. 2a,b). The NCAM $^+$ levels increased until day 10 to >80% at 7.5/3, 15/1 and 15/3 but remained <20% at 7.5/1 (Supplementary Fig. 2a).

cKIT $^+$/CXCR4 $^+$ expression (characteristic of endodermal progenitors[22]) revealed a reversed pattern, namely relatively high proportion of 28 ± 12% double-positive cells at 7.5/1 and nearly absence (0.67 ± 0.31%) at 15/3; cardio-inductive conditions 15/1 and 7.5/3 showed intermediate levels of 9.4 ± 1.8% and 11.1 ± 3.4%, respectively (Fig. 2c).

Thus, cardiogenic cornerstones showed a similar expression pattern of early mesendoderm progenitors, while cells at non-cardiogenic settings were primed into opposing directions either typical of definitive endoderm (primed anterior to PCM along the PS) in 7.5/1 or of PSM (specified posterior to PCM) in 15/3.

**Cornerstone-specific PS patterns are cell line-independent.** Flow cytometry directly after CHIR treatment revealed BCD-dependent expression patterns of the PS markers T brachyury (T) and MIX1 homeobox-like protein 1 (MIXL1). Using a MIXL1-GFP reporter line[23], distinct expression in 2D and 3D was found, reflecting NCAM patterns on day 3 with significantly higher MIXL1-GFP $^+$ at 15/3 (76.7 ± 2.4%) but significantly lower levels at 7.5/1 (16.01 ± 1.8%) as compared with 15/1 (56.8 ± 1.6%) and 7.5/3 (48.72 ± 2.7%; Fig. 2d,e). Equivalent patterns were observed for T applying the NKX2.5-GFP- (Fig. 2f,g) and four different human induced pluripotent stem cell lines (Supplementary Fig. 2b) established by various

reprogramming technologies (Supplementary Methods). This confirms manifestation of cornerstone-specific, cell line-independent expression of PS markers readily at 24 h of differentiation.

**BCD predominates CHIR in global gene expression patterns.** Microarray analyses of cornerstone conditions and controls (undifferentiated cells and CHIR-free differentiation) at 24 h were conducted. Principal component analysis revealed clear

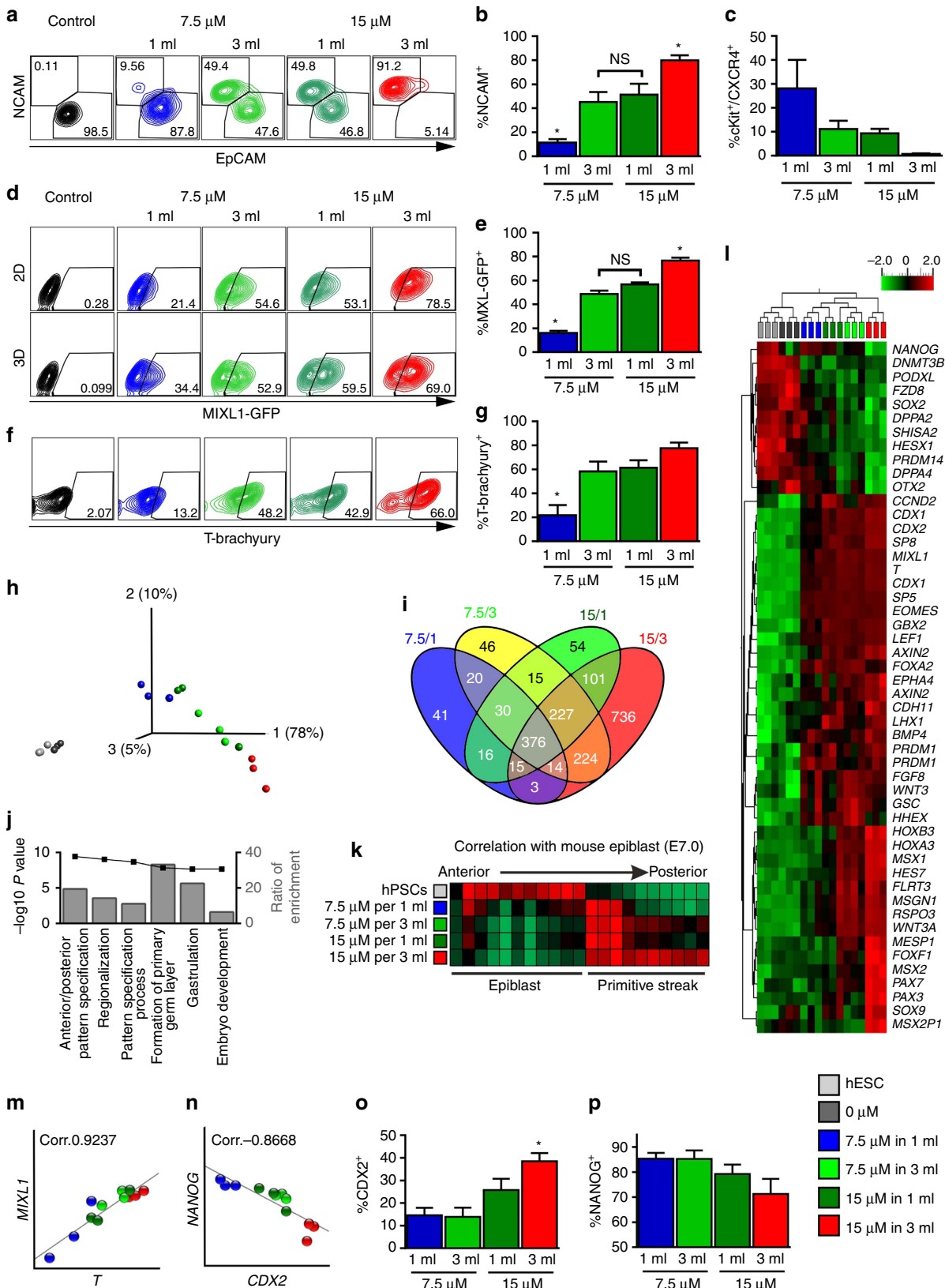

separation of CHIR-treated versus control conditions (Fig. 2h). Within the CHIR-treated group, cardio-inductive conditions (light and dark green) did not spread into separate groups but intermingled between the other extremes (blue and red).This highlights manifestation of distinct global transcriptome patterns readily after 24 h. Remarkably, Venn diagram analysis of >2-fold regulated genes (of treated compared with undifferentiated hPSCs) revealed only 15 genes that were exclusive to cardio-inductive conditions (Fig. 2i). Additional statistical testing ($P < 0.01$) showed only a single gene (LOC100133311) of significant differential expression at cardiogenic conditions. This suggests that PCM is mainly specified by an intermediate expression pattern along the PS rather than defined by an exclusive set of upregulated lineage specifiers, in line with recently published spatially resolved transcriptomic data sets of the PS state in mice and chicken embryos[24,25] (Supplementary Fig. 2c,d).

The Venn diagram depicts a distinct overlap of 224 genes between 7.5/3 and 15/3 conditions (representing BCD dependence) compared with 101 genes exclusive to the overlap of 15/1 and 15/3 (representing CHIR dependence). This was confirmed by statistical testing ($P < 0.01$) revealing 105 low BCD (that is, high medium volume) versus 18 high CHIR-regulated genes. This highlights the notion that the BCD effect dominates the impact of the CHIR concentration at our experimental conditions. Gene ontology analysis of >5-fold regulated genes showed a clear association with early PS patterning, including anterior/posterior specification (Fig. 2j). Unbiased regional allocation of each condition to the mouse epibast (E7.0) by zipcode mapping[24] confirmed an epiblast-like stage of the undifferentiated cells and increasing anteroposterior progression along the PS axis at increasing CHIR and medium volume conditions.

Hierarchical clustering of genes association with pluripotency, PS formation and gastrulation (Fig. 2l) revealed the close relation of 7.5/1 (blue) and 15/1 (dark green), further underlining that the BCD predominates the CHIR concentration. The distant-most pattern to undifferentiated cells at 15/3 was accompanied by the most prominent downregulation of pluripotency markers, including NANOG, SOX2, DPPA2 and DPPA4 (Fig. 2l). Conversely, markers of PS progression MIXL1 and T were upregulated most at 15/3 confirming flow cytometry results (Fig. 2f,g) by the high level of MIXL1 and T correlation in individual samples on transcriptional level (Fig. 2m). However, 15/3 conditions were also unique regarding a number of upregulated genes, including MSGN1, PAX3, PAX7, HOXA3, HOXB3, AXIN2, CDX1 and CDX2 (specific to PSM/paraxial mesoderm[26–28]), as well as CCND2, SP5, SP8, MSX1, MSX2, SOX9, PAX3 and PAX7 (markers of posterior neural plate border formation[26,29], Fig. 2l and Supplementary Fig. 2e displaying HOX gene pattern). Upregulation of FOXF1 and LHX1 further suggests

a close relation of 15/3-primed cells to a lateral plate mesoderm-like fate, which, in the embryo, is located posterior to cardiac and anterior to paraxial mesoderm along the PS[30,31].

Low expression of endoderm-related markers OTX2, HHEX and SHISA2 at 15/3 reflects the absence of cKIT+/CXCR4+ (Fig. 2c). Highest NANOG expression at 7.5/1 negatively correlated with CDX2 expression separating 7.5/1 against 15/3; this, again, is intermingled by 7.5/3 and 15/1 (Fig. 2n–p) confirming anteroposterior mesoderm patterning by NANOG/CDX2 (ref. 28).

**WNT activity is BCD-sensitive at 7.5 but not 15 μM CHIR.** Following confirmation of CHIR stability over 48 h under experimental conditions (Supplementary Fig. 4a), multiplex arrays were applied for detailed status analysis of the WNT pathway effector β-catenin[32,33]. A first response was readily detected after 2 h (120 min) post CHIR exposure (Supplementary Fig. 4b), preceding end-point patterns observed at 24 h (1,440 min; Supplementary Fig. 4b). Continuous increase in signal activity interestingly suggests diminished negative feedback of the WNT pathway in the presence of CHIR (Supplementary Fig. 4b) confirming other cell systems[32,34].

Statistical analysis revealed minor impact of the BCD on the β-catenin status at 15 μM CHIR (Fig. 3a–h) suggesting that differential pattering of 15/1 versus 15/3 is largely independent of WNT pathway activity. In contrast, comparison of 7.5/1 and 7.5/3 revealed significantly higher β-catenin activity at 7.5/3 (low BCD; Fig. 3). This suggests that canonical WNT pathway activity is more sensitive to BCD-dependent feedback at lower CHIR concentrations.

**Differential accumulation of secreted factors after 24 h.** Since CHIR-dependent β-catenin activity alone cannot explain the differentiation results, we postulated a role for CHIR/BCD-dependent accumulation of secreted pathway modulators. This idea is further suggested by the cornerstone-specific gene expression patterns obtained for key paracrine modulators of development (Supplementary Fig. 5a). To test the hypothesis that differential secretomes direct PS patterning in our system, we exchanged the entire medium after 6 h of CHIR treatment, but maintaining respective CHIR/BCD conditions. This indeed altered cardiomyogenesis at day 10, stressing the assertive impact of secreted factors at very early differentiation stages (Fig. 4a). At 7.5/1, NKX2.5-GFP+ increased from $2.4 \pm 0.8$ to $39.0 \pm 5.0\%$ (Fig. 4b). Conversely, at cardio-inductive conditions 7.5/3 and 15/1, GFP+ expression significantly decreased from >50 to <25%; yet at 15/3, GFP+ remained at 13–14%.

**Figure 2 | Distinct mesendodermal identities are manifested within 24 h of differentiation.** (**a**) Representative density plots for NCAM/EpCAM and (**b**) analysis of NCAM+ of eight independent experiments on day 3. *$P < 0.05$, evaluated by one-way analysis of variance (ANOVA) with *post hoc* Bonferroni analysis. (**c**) Flow cytometric analysis for cKIT+/CXCR4+ of three independent experiments on day 3 ($n = 3$ independent experiments). (**d,e**) Representative density plots showing MIXL1-GFP on day 1 in 2D monolayer (upper panel), 3D suspension (lower panel) and respective quantification in $n = 5$ independent experiments. *$P < 0.001$, evaluated by one-way ANOVA with *post hoc* Bonferroni analysis. (**f,g**) Representative density plots showing T-brachyury on day 1 from suspension-based differentiation and respective quantification ($n = 5$ independent experiments). *$P < 0.05$, evaluated by one-way ANOVA with *post hoc* Bonferroni analysis. (**h**) Principal component analysis of microarray data. Each dot represents an independent sample collected after 24 h of differentiation and undifferentiated hESCs. (**i**) Venn diagram of >2-fold regulated genes in the four conditions after 24 h compared with undifferentiated cells. (**j**) Top-ranked gene ontology terms without pre-selection[58] associated with >5-fold regulated gens in the four conditions. (**k**) Spatial allocation of the each cornerstone condition to the mouse epiblast (E7.0) based on zipcode mapping of whole-transcriptome data along the primitive streak. Red = high correlation; green = low correlation. (**l**) Heatmap of differentially regulated genes ($P < 0.001$) associated with pluripotency, primitive streak, early endoderm, early mesoderm and neural plate border. (**m,n**) Correlation of MIXL1/T-brachyury and NANOG/CDX2 in CHIR-treated conditions. (**o,p**) Flow cytometric analysis of CDX2 and NANOG on day 1. $n = 5$ and 3 of independent experiments, respectively. *$P < 0.01$ compared with 7.5 μM conditions evaluated by one-way ANOVA with *post hoc* Bonferroni analysis. All bars shown in this figure represent mean ± s.e.m. See also Supplementary Figs 2 and 3. NS, not significant.

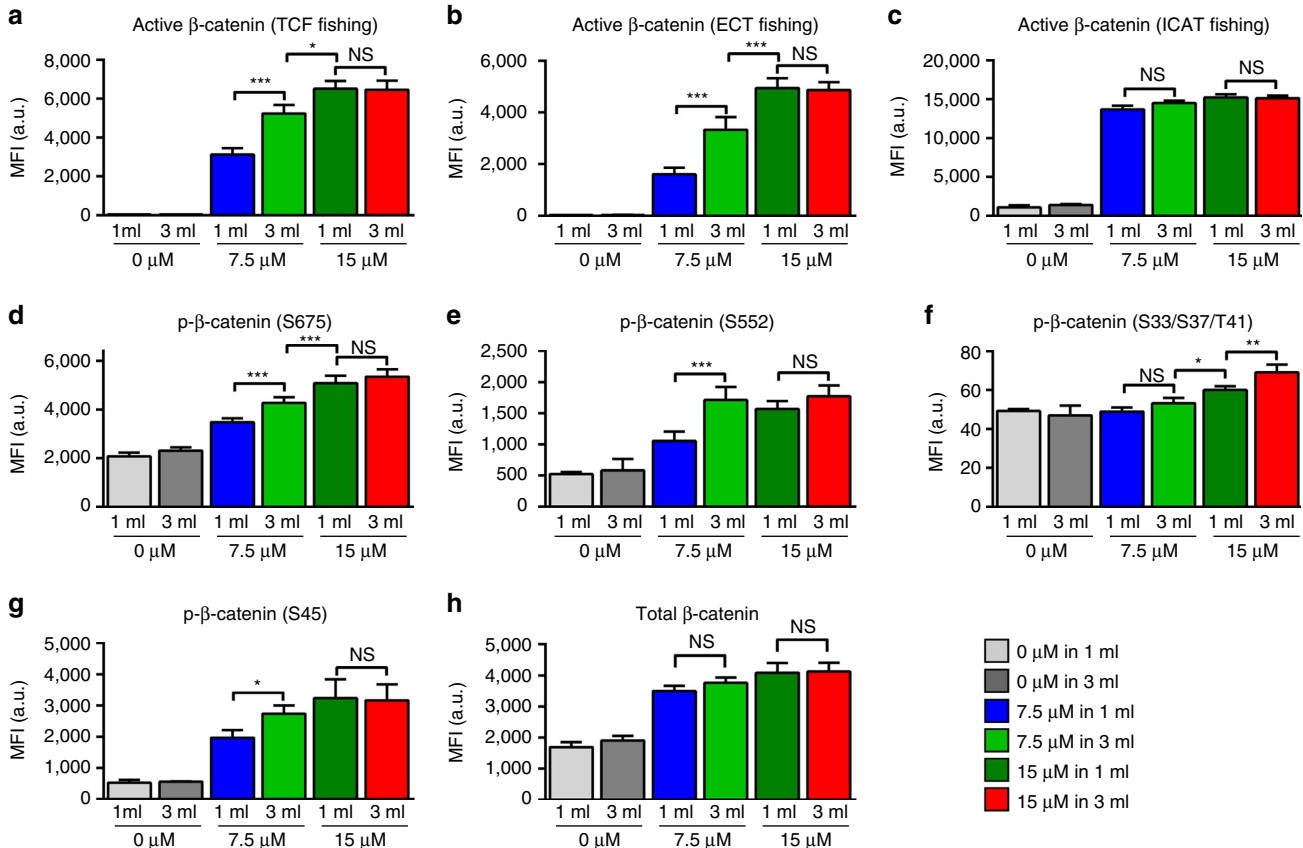

**Figure 3 | β-catenin activity is volume-dependent at 7.5 μM but not at 15 μM CHIR.** (a–c) Protein–protein interaction assay for transcriptionally active β-catenin after 24 h of differentiation using glutathione S-transferase-tagged T-cell factor (TCF), cytosolic tail of E-cadherin (ECT) and inhibitor of beta-catenin and TCF-4 (ICAT). (d–h) Sandwich immunoassay detecting total β-catenin and phosphorylation at S675, S552, S33/S37/T41 and S45 after 24 h of differentiation. n = 9 from 3 independent experiments of 3 biological replicates each. *P < 0.05, **P < 0.01, ***P < 0.001, evaluated for each target by one-way analysis of variance with post hoc Bonferroni analysis. All bars shown in this figure represent mean ± s.e.m. See also Supplementary Fig. 4. NS, not significant.

Supernatant analysis of secreted WNTs was exemplary performed by a quantitative protein array (exemplarily shown for DKK1 in Supplementary Fig. 5b) and confirmed by western blot (Supplementary Fig. 5c). Strikingly >6-fold higher concentrations of the WNT antagonist Dickkopf-1 (DKK1) was detected in 3 ml (low BCD) compared with respective 1 ml settings (Supplementary Fig. 5a–c). Accounting for the medium volume >18-fold higher DKK1 accumulation was found at 7.5/3 and 15/3 (low BCD) compared with 7.5/1 and 15/1, matching respective gene expression data (Supplementary Fig. 5b, right). In light of this, it was surprising that exogenous DKK1 supplementation parallel to CHIR had no apparent impact on cardiomyogenesis at any cornerstone condition (Fig. 4c and Supplementary Fig. 5d). In contrast, the chemical DKK1 inhibitor WAY-262611 (ref. 35) significantly reduced NKX2.5-GFP+ levels (Supplementary Fig. 5e,f). Assuming specificity of WAY-262611, these findings might suggest that endogenous DKK1, per se, is required for proper mesendoderm formation/specification in hPSCs equivalent to early embryogenesis[36] but supplementation of recombinant DKK was ineffective in modulation differentiation in our system (Fig. 4c and Supplementary Fig. 5d).

Expression of SFRP1, another secreted WNT antagonist, showed a reversed pattern compared with DKK1, that is, higher gene and protein levels at 1 versus 3 ml (Supplementary Fig. 5g,h), whereby highest SFRP1 levels at 7.5/1 and at non-CHIR controls suggests its repression along CHIR-triggered

(posterior) progression of differentiation (Supplementary Fig. 5a,g,h). However, recombinant SFRP supplementation showed a minor effect in our experiments (Fig. 4c) equivalent to DKK1 addition.

**BMP pathway modulation provokes anteroposterior shifts.** Testing robustness of the cornerstone conditions to other established modulators of hPSC differentiation and cardiomyogenesis revealed remarkable stability to activin A, insulin or basic fibroblast growth factor (bFGF) supplementation (Fig. 4c). In contrast, BMP4 (but not BMP2) significantly altered differentiation (Fig. 4c–e), notably reflecting patterns of additional medium exchange (Fig. 4a,b): NKX2.5-GFP+ increased at 7.5/1 (from 8.0 ± 4.1 to 56.5 ± 4.5%) but dropped at 7.5/3 and 15/1 (from 57.0 ± 4.4% to 34.0 ± 8.0% and 56.8 ± 4.1% to 22.5 ± 4.5%, respectively; Fig. 4e). Combined with findings that MIXL1-GFP+ increased across all cornerstones (Fig. 4f), overall posteriorization by BMP4 is strongly suggested by this data.

BMP4 addition hardly perturbed the WNT pathway at the level of β-catenin activity (Supplementary Fig. 6) but some reduction was observed at 15/1 conditions suggesting a direct crosstalk between the BMP and WNT pathways as previously published[37]. Opposing BMP signalling by dorsomorphin[38] resulted in significant NKX2.5-GFP reduction at 7.5/3 (Fig. 4g), suggesting cornerstone conditions' dependency on endogenous BMP signalling and subsequent anteriorization towards 7.5/1-like

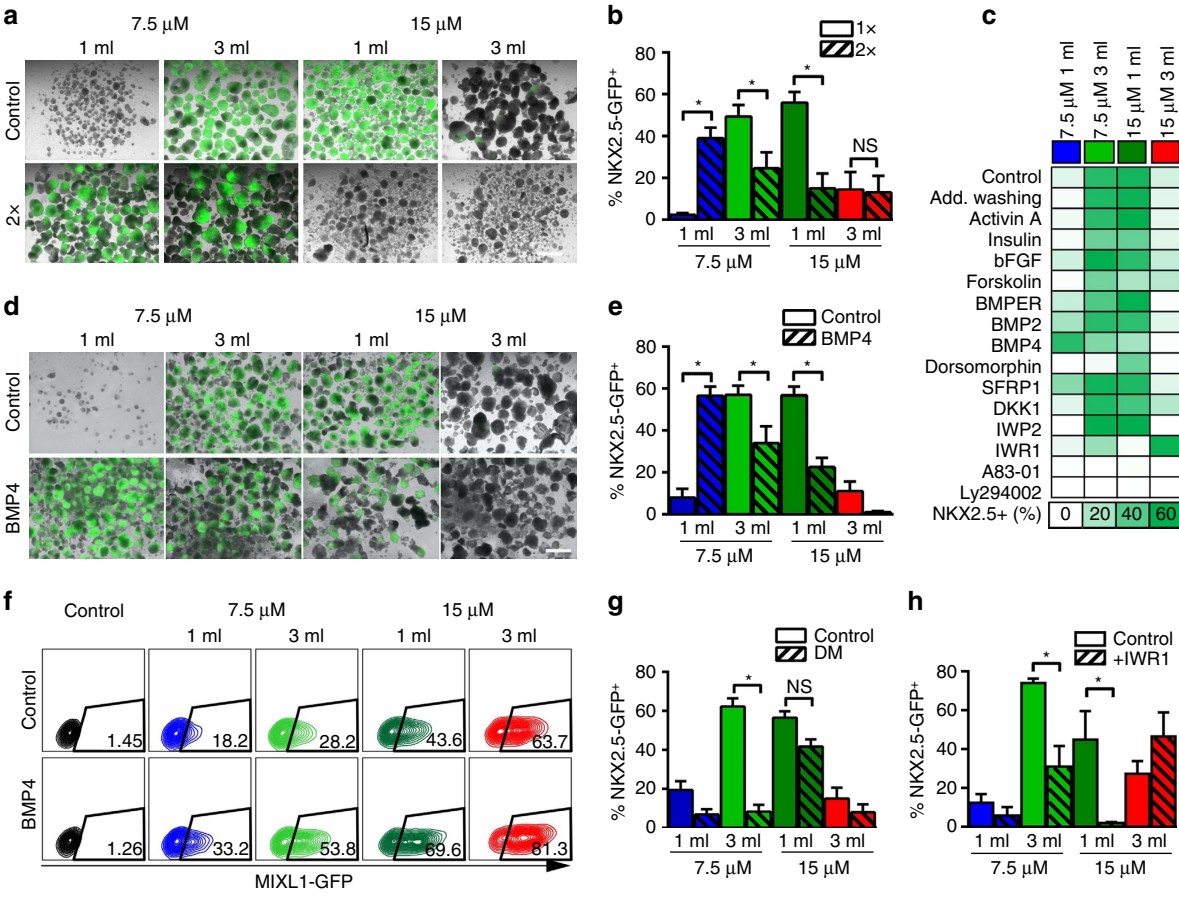

**Figure 4 | A distinct balance of WNT and BMP signalling is required for cardiomyogenesis.** (**a**) Representative images showing the effect of additional medium refreshment after 6 h during the 24 h CHIR treatment on the NKX2.5-GFP expression measured on day 10 and (**b**) the analysis of six independent experiments thereof. (**c**) Heatmap showing induction of NKX2.5-GFP$^+$ on day 10 following concomitant addition of indicated substances during the 24 h CHIR treatment. (**d**,**e**) Representative images on day 10 after concomitant addition of BMP4 to CHIR treatment and flow cytometric analysis thereof ($n = 3$). (**f**) Representative flow cytometric data showing increase in MIXL1$^+$ following concomitant BMP4 treatment after 24 h. (**g**,**h**) Effect of concomitant addition of either dorsomorphin (DM) (**g**) or IWR1 (**h**) to CHIR treatment on NKX2.5-GFP$^+$ determined by flow cytometry on day 10 ($n = 4$ of independent experiments in both graphs). All bars shown in this figure represent mean ± s.e.m. *$P < 0.05$, evaluated by one-way analysis of variance with *post hoc* Bonferroni analysis in all bar graphs. Scale bars, 1 mm. See also Supplementary Figs 5 and 6. NS, not significant.

conditions by dorsomorphin. Notably, dorsomorphin only marginally affected 15/1 (Fig. 4g), which is opposed by the significant impact of BMP4 supplementation outlined above (Fig. 4e).

BMP and WNT signalling (conducted through SMAD 1/5/8 and ß-catenin, respectively) converge in controlling mesendoderm specification on the transcriptional level[28,39]. Since antagonizing BMP by dorsomorphin hardly affected the 15/1 and 15/3 conditions, we have postulated their predominant WNT dependence. Indeed, direct inhibition of WNT pathway activity at the ß-catenin level by IWR1 supplementation (that is, by promoting ß-catenin degradation via stabilization of AXIN) increased NKX2.5-GFP$^+$ levels at 15/3 (Fig. 4h) suggesting expected cornerstone anteriorization towards PCM (that is, adjustment of 15/3 towards 7.5/3-like conditions). Accordingly, IWR1 disrupted cardiomyogenesis at 15/1, supposedly by anteriorization towards 7.5/1-like conditions. Notably, supplementation of the WNT inhibitor IWP2, a chemical inhibitor of PORCUPINE, which acts by disrupting secretion of WNTs, did not alter the cardiac differentiation outcome (Supplementary Fig. 5i). This notably suggests a minor role of secreted WNTs in specifying cornerstone conditions, in line with the vastly ineffective supplementation of DKK1 or SFRP1.

Together our results confirm the established complexity of the WNT and BMP signalling pathways' interplay in defining cells' distinct positioning along the PS[28,39]. At cornerstone conditions, lacking extrinsic BMP supplementation, BMP pathway activity is a result of both the CHIR concentration and the BCD; vice versa BMP activity shapes PS patterning (that is, promotes posteriorization) in a BCD-dependent manner, including feedback on the CHIR-controlled WNT pathway activity.

**Secreted LEFTY1 and CER1 restrict posterior PS progression.** To further investigate BCD effects and their time dependence, supernatants were collected at 6 or 24 h from 7.5/1 conditions (most anterior PS specification). Purified proteins were supplemented to 15/3 (most posterior) parallel to CHIR exposure (Fig. 5a). The '6 h secretome' elevated NKX2.5-GFP$^+$ from 11.1 ± 2.5 to 38.6 ± 4.2%, suggesting the expected presence of anteriorizing factors. This anteriorizing activity was surprisingly reduced when supplementing proteins from 24 h supernatants (Fig. 5a), suggesting a shifted balance between anteriorizing and posteriorizing factors in the supernatant collected at the later stage.

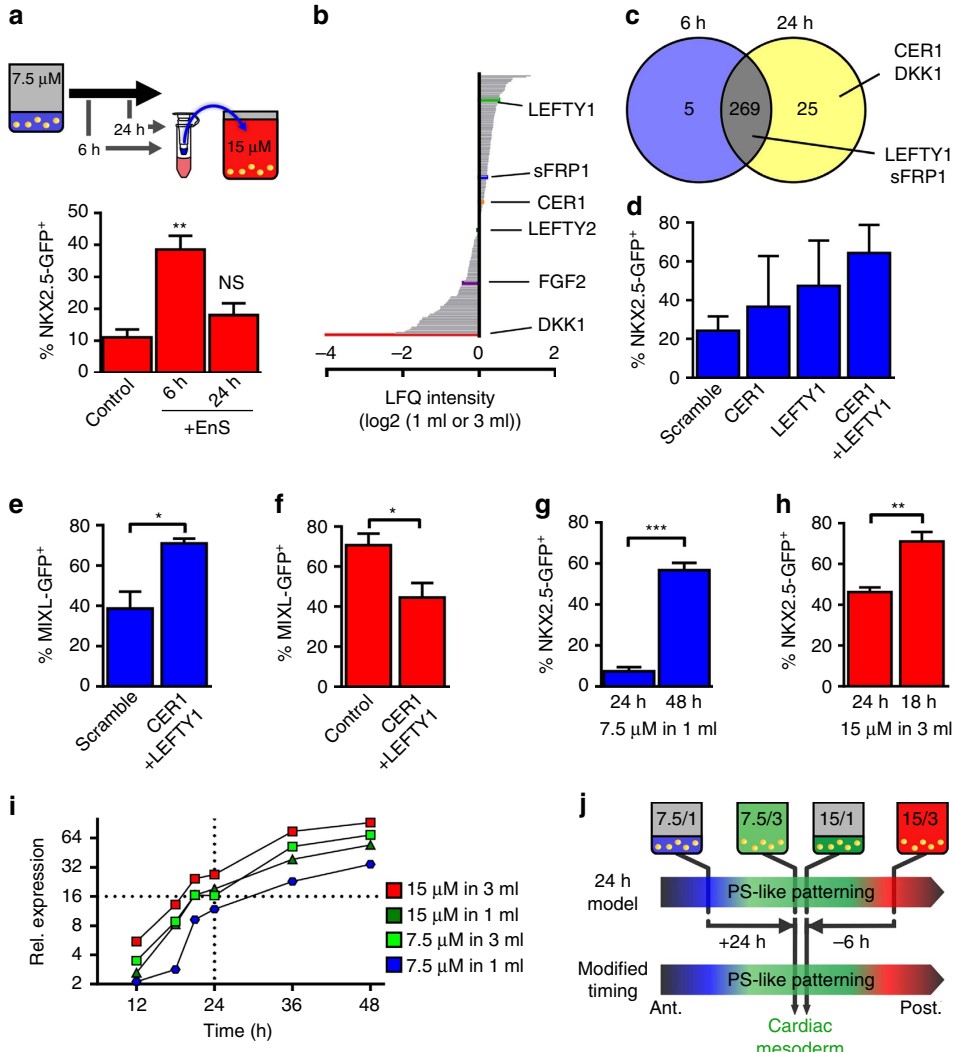

**Figure 5 | Paracrine LEFTY1 and CER1 suppress differentiation progression.** (**a**) Enriched supernatant (EnS) collected after 6 and 24 h from cells cultured at 7.5/1 was added for the first 24 h to the differentiation conducted at 15/3 (upper panel). The impact of the supernatant on emergence of NKX2.5-GFP$^+$ was determined by flow cytometry on day 10 (lower panel; $n = 6$ independent experiments). **$P < 0.01$ compared with control, evaluated by one-way analysis of variance with *post hoc* Bonferroni analysis. (**b**) Quantification of proteins detected in the supernatant conditioned at 7.5/1 7.5/3 revealed high levels of LEFTY1 in 1 ml and high levels of DKK1 in 3 ml. (**c**) Venn diagram of detected proteins in the supernatant conditioned for 6 and 24 h confirms early presence of LEFTY1 and SFRP1, and induction of CER1 and DKK1 at later stage. (**d**) siRNA knockdown of CER1 and LEFTY1 before differentiation at 7.5/1 resulted in increased levels of NKX2.5-GFP$^+$ expression ($n = 4$ of independent experiments). (**e**) Dual-knockout of CER1 and LEFTY1 increased the MIXL1-GFP$^+$ after 24 h at 7.5/1. $n = 3$ of independent experiments. *$P < 0.05$, evaluated by Student's $t$-test. (**f**) Combined addition of recombinant CER1 (10 μg ml$^{-1}$) and LEFTY1 (100 ng ml$^{-1}$) decreased the MIXL1-GFP$^+$ after 24 h at 15/3. $n = 3$ of independent experiments. *$P < 0.05$, evaluated by Student's $t$-test. (**g**) Increasing CHIR exposure from 24 to 48 h at 7.5/1 resulted in increased NKX2.5-GFP$^+$ expression as determined by flow cytometry on day 10. $n = 6$ of independent experiments. ***$P < 0.01$, evaluated by Student's $t$-test. (**h**) Decreasing CHIR exposure from 24 to 18 h at 15/3 resulted in increased NKX2.5-GFP$^+$ expression as determined by flow cytometry on day 10. $n = 4$ of independent experiments. **$P < 0.01$, evaluated by Student's $t$-test. (**i**) Time course analysis of MIXL1 induction during the first 24 h in the four conditions shown as mean of $n = 3$ independent experiments. (**j**) Model of primitive streak patterning by modulation of CHIR exposure. All bar graphs shown in this figure represent mean ± s.e.m. See also Supplementary Fig. 5j. Ant, anterior; NS, not significant; Post, posterior.

Secretome analysis from 7.5/1 and 7.5/3 by tandem mass spectrometry (MS/MS) identified 269 proteins allocated to the gene ontology term 'extracellular space' (Fig. 5c, entire list of proteins can be found as Supplementary Data 1). DKK1 was the highest enriched factor in 7.5/3 (Fig. 5b), confirming the protein array and western blotting data (Supplementary Fig. 5b,c). Notably, LEFTY1, a component of NODAL signalling, was detected among the highest-enriched proteins at 7.5/1 (Fig. 5b) and readily detectable in the 6 h secretome (Fig. 5c). In all, 25 proteins were '24 h-secretome exclusive', including CER1 (Fig. 5c), consistent with CER1 upregulation in response to CHIR (Supplementary Fig. 5a).

Since CER1 and LEFTY1 were described to restrict posterior progression of PS in mouse embryos[40], we tested their knockdown at most anteriorizing conditions, the 7.5/1 cornerstone. Compared with scrambled siRNA treatment, a 2.6-fold increase in NKX2.5-GFP$^+$ (from 24.3 ± 7.5 to 64.4 ± 14.5%; Fig. 5d) was observed for combined knockdowns, whereby single siRNA treatments resulted in partial GFP upregulation. In line with this, the dual knockdown resulted in an increase of MIXL1$^+$ from 38.7 ± 8.4 to 71.0 ± 2.3% after 24 h of CHIR treatment (Fig. 5e). Vice versa, differentiation progression was anteriorized by concomitant addition of recombinant LEFTY1 and CER1 to the

most posterior condition (15/3) as indicated by the decrease of MIXL1$^+$ after 24 h from 70.8 ± 5.6 to 44.5 ± 7.3% (Fig. 5f). This was further supported by partial rescue of the cardiac differentiation at 15/3 by addition of LEFTY1 or CER1, respectively (Supplementary Fig. 5j).

Together, this suggests a decisive role of LEFTY1 and CER1 in the paracrine milieu, which is accountable for the anteriorizing activity of high BCD, thereby delaying PS progression at 7.5/1 and 15/1.

**Duration of CHIR/BCD treatment controls PS progression**. To investigate the time-dependence in this process, CHIR/BCD treatment was prolonged from 24 to 48 h at 7.5/1. Significant NKX2.5-GFP$^+$ increase from 7.4 ± 2.0 to 56.8 ± 3.5% (Fig. 5g) suggests PS progression by prolonged incubation, thereby apparently shifting definitive endoderm specification of 7.5/1 after 24 h towards cardiomyogenesis after 48 h. Conversely, reducing CHIR/BCD incubation from 24 to 18 h at 15/3 increased NKX2.5-GFP$^+$ from 46.3 ± 2.3 to 71.08 ± 4.7% (Fig. 5h). This finding correlates with time-dependent reduction of PS progression at 15/3 from PSM towards the anterior PCM. To substantiate the time-dependent context, MIXL1-GFP expression, representing an informative marker of PS progression (Fig. 2d,e), was monitored over 48 h at all four cornerstones. Condition-specific expression kinetics (Fig. 5i) revealed that continuous MIXL1 upregulation occurred at all conditions, suggesting that in principle any distinct PS positioning can be achieved at any cornerstone by respective timing. Relative MILX1 levels specific to 7.5/3 and 15/1 at 24 h (representative of PCM specification in our experiments) are readily achieved within ∼18 h at 15/3 but only after >30 h at 7.5/1. These findings, which are in good correlation with the independent cardiomyogenic analysis in Fig. 5g,h, are summarized in a resulting model in Fig. 5j.

## Discussion

By kick-starting the process only by CHIR, recent studies have established directed hPSC differentiation into multiple lineages, including definitive endoderm[41,42] and definitive endoderm-derived hepatocytes[43], PCM-derived cardiomyocytes[7,17,18,44] and endothelial progenitors[45], as well as PSM-derived chondrocytes[28] and skeletal myoblast progenitors[27]. Moreover, efficient cardiomyogenesis was achieved at CHIR concentrations ranging from 5 to 15 μM (refs 7,17,18,44), notably overlapping with 3–5 μM used for definitive endoderm[41–43] and 6–10 μM for PSM derivatives[27,28].

But how can glycogen synthase kinase 3 beta (GSK3ß) inhibition determine such lineage variety, yet at broadly overlapping CHIR concentrations? Applying CHIR, Mendjan et al.[28] showed that hPSCs are not turned into multipotent PS progenitors maintaining prolonged lineage plasticity. Instead, a restricted PS model was favoured suggesting that cells are readily primed into restricted fates along a PS-like axis within 48 h of differentiation[28]. Here, using cardiogenesis as central reference point of anteroposterior specification, we confirm the restricted PS model notably showing that distinct cell priming is readily established at 24 h of CHIR treatment; at the secretome level decisive conditions are present at 6 h and differential ß-catenin activity can be measured as early as 2–3 h post CHIR supplementation (Supplementary Fig. 4).

One key aim of our study was to conclusively elucidate how the BCD deflects CHIR-induced hPSC priming. We show that increasing CHIR progressively pushes PS priming towards posterior fates; omitting CHIR leaves cells 'uninstructed' and fading. In contrast, elevating the BCD restricts PS progression. Thus, at any given CHIR dose and incubation time, relatively

higher BCD results in more anterior cell fate determination. This raises the question of how this effect is mediated.

Extra media changes and swapping enriched protein samples from the supernatants of opposing cornerstone conditions strongly suggest that the BCD effect is mediated by paracrine factors. Despite the overall complexity of these secretomes, we showed that accumulation of individual factors is highly differential and not merely following a simple linear correlation with the BCD.

Two observations forced our attention on released factors expressed in hPSCs ahead of differentiation. First, medium refreshment after 6 h induced posteriorization suggesting very early accumulation of anteriorizing factor(s) depleted by the medium change. Second, secreted factors from 7.5/1 (high BCD, most anteriorizing conditions) collected at 6 h resulted in anteriorization of 15/3 (most posteriorizing condition). In contrast, the 7.5/1 secretome collected at 24 h did not induce anteriorization. This confirms the early accumulation of anteriorizing factors at high BCD. Moreover, it highlights the dynamic changes of the paracrine milieu not only in between different cornerstone conditions but also within the same condition throughout the first 24 h.

Secretome analysis revealed that the NODAL signalling antagonists, TGFß family member LEFTY1 was among the most prominently accumulated proteins at high BCD and readily detected in secretome samples at 6 h (Fig. 5b,c). LEFTY1 knockdown resulted in posteriorization of the 7.5/1 cornerstone (Fig. 5d), suggesting that LEFTY1 accumulation is a key source of anteriorizing activity along rising BCD, thereby antagonizing progression of CHIR-triggered differentiation. Recently, single-cell RNA sequencing of the human blastocyst revealed that LEFTY1, together with other key components of the TGFß signalling, is enriched in the human epiblast[46]. TGFß signalling was found to control NANOG expression in epiblast cells revealing pathways' requirement in pluripotency, whereby conservation of pluripotency control between the human embryo and hESC was highlighted[46]. However, secretome analysis in our study confirmed accumulation of another NODAL antagonist, CER1, at high BCD conditions in 24 h secretome samples comparing 7.5/1 versus 7.5/3 (Fig. 5b,c) in line with factors' upregulation at the RNA level (Supplementary Fig. 5). Equivalent to LEFTY1, knockdown of CER1 also resulted in posteriorization, which was additive to the effect of LEFTY1 reduction in double-knockdown experiments (Fig. 5d). Besser[47] demonstrated regulatory crosstalk between WNT and TGFß signalling in hESCs. The study suggests that secreted WNTs downstream of GSK3ß inhibition upregulate NODAL signalling, inducing activation of the ALK4/5/7 and SMAD 2/3 branch of the pathway, notably counteracted by LEFTY1 and two proteins. Moreover, findings by Besser suggest CER1 as a potential inhibitor of the SMAD 1/5/8 branch of the TGFß pathway, thus counteracting pathways' activation via secreted BMP agonists. By these mechanisms LEFTIES and CER1 act in parallel and in conjunction as retarder of hESC differentiation[47]. This is in good agreement with our results at the earliest steps of differentiation, showing decelerated progression of posterior PS priming by LEFTY1 and CER1 supplementation (Fig. 5f), likely acting by antagonizing the accelerating signals of WNT and BMP pathway agonists.

Interestingly, a study by Hough et al.[48] identified CER1 and LEFTY1 as key candidates contributing to the heterogeneity in hPSC cultures, which is another hint regarding their pivotal role at the edge of pluripotency versus differentiation.

Our study has substantial practical consequences. It provides a plausible, mechanistic explanation for how a specific CHIR concentration, without co-supplementation of other factors, can

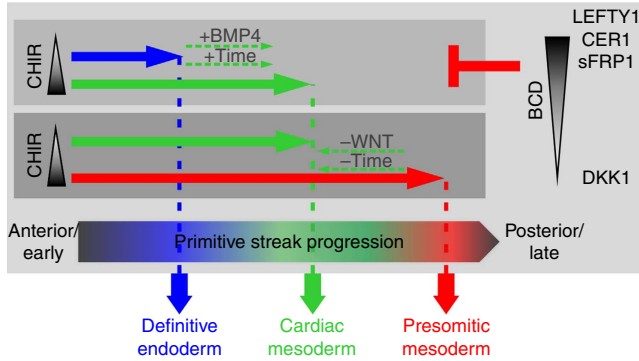

**Figure 6 | Summary of decisive factors in PS patterning revealed in this study.** The four cornerstone conditions investigated in this study result in distinct progression of the hPSCs during exit of pluripotency. Depending on the level of CHIR (WNT pathway) stimulation and the BCD, primitive streak-like patterning along the anteroposterior axis occurs that ultimately determine cell fate into prospective definitive endoderm, cardiac mesoderm and PSM. WNT (CHIR) and BMP signalling act as main driver of PS progression over time, while cell-borne factors (LEFTY1 and CER1) antagonize the progression of the streak formation depending on the BCDs.

induce hPSC differentiation into entirely different lineages by means of BCD and time modulation, as graphically depicted in Fig. 6. Reverse, we highlight the utmost importance of providing detailed experimental information on the BCD, which is usually not accounted for in most publications. Notably, despite showing relevance to any culture format, using a monolayer-based expansion/differentiation platform might be less sensitive regarding the BCD. Irrespective of the seeded cell number, relatively similar cell densities are typically achieved (per well or flask) after some days of culture in 2D, simply due to surface restrictions. Subsequently, by applying a defined medium volume for differentiation and closely monitoring timing of CHIR exposure, inter-experimental and trans-laboratory reproducibility in 2D is facilitated; but edge-effects at the rim of a well or flask have been noted by us and others[49].

The consequences for hPSC culture and differentiation in 3D suspension are considerably more drastic. Suspension culture technology strongly facilitates process scale-up, including transition into stirred bioreactors[4,50–53]. However, control of the cell density is more critical in 3D since no surface-limited restriction exist. Moreover, we showed that transition from monolayer to matrix-free hPSC suspension culture is accompanied by district WNT pathway modulations at the interplay of ß-catenin and E-cadherin[54]. In addition, elevated expression of LEFTY1 at the gene and protein level was observed without notable impact on pluripotency[54]. In light of the current study, such culture platform-dependent cellular and molecular changes suggest mayor adaptation requirements for the transition of directed lineage differentiation protocols from 2D to 3D. Furthermore, modulating hPSC aggregate formation in 3D ahead of differentiation, for example, by the inoculation density, stirring conditions and process feeding patterns, likely shapes the secretome; variations of which may underlie variations in lineage differentiation, which we have recently postulated in a study on hPSC-cardiomyocyte production in a stirred bioreactor[17].

Another important prediction from the present study is that to ensure successful differentiation into a desired lineage at higher cell density, for example, to improve cell yields and process efficiency in a restricted culture format, will require a higher CHIR concentration when the incubation time is fixed. Notably, however, incubation timing becomes even more critical at higher CHIR concentrations, as it accelerates progression of PS priming.

In summary, our study provides a novel mechanistic and predictive understanding of the earliest stages of mesendodermal priming of hPSCs, promotes better experimental control of these processes and supports more rational translation into alternative culture conditions.

## Methods

**hPSC culture.** HES3 NKX2-5[eGFP/w] (ref. 19), HES3 MIXL[eGFP/w] (ref. 23), hCBiPS2 (CBiPS)[55], HSC_F1285_T-iPS2 (HSCiPS)[49] and hHSC_F1488_SeV-iPS2 (nt-iPS; this study) were maintained under standard conditions on irradiated mouse embryonic fibroblasts (MEFs). hHSC_Iso4_ADCF_SeV-iPS2 (animal-derived component-free conditions (ADCF) nt-iPS; this study) were maintained in E8 medium on Synthemax-coated culture plates. Before differentiation, cells were expanded for at least three passages under feeder-free conditions on geltrex-coated tissue culture flasks (TPP) in MEF-conditioned medium, mTeSR1 (Stem Cell Technologies) or E8, and passaged at $5 \times 10^4 \, \text{cm}^{-2}$ using accutase (PAA). Cell lines have been karyotyped and were regularly tested for mycoplasma using MycoAlert detection kit (Lonza).

**Generation and characterization of non-transgenic iPSCs without or with animal-derived component-free conditions.** A total of 200,000 cord blood CD34 + haematopoietic stem cells (either kindly provided by Vita34 or isolated and cultivated under animal-component-free conditions) were transduced with CytoTuneTM-iPS Sendai Reprogramming Kit (Invitrogen) at 37 °C for 24 h in haematopoietic stem cell medium or ADCF-haematopoietic stem cell medium (IMDM, 10% fetal bovine serum, 100 ng ml⁻¹ interleukin-3, SCF, Flt-3 and TPO; all from Peprotec) or Stem-Span-ACF-Medium (Stem Cell Technologies; containing: $8 \, \mu\text{g ml}^{-1}$ polybrene). To reduce residual virus vectors, cells were centrifuged and reseeded in fresh medium the next day. Cells were transferred onto mitotically inactivated MEFs or Corning Synthemax II-SC Subtrate (Corning)-coated culture plates 4 days post transduction. On day 5, medium was changed to PSC medium containing Knockout DMEM, serum replacement (20%), non-essential amino acids (1%), L-glutamine (0,1%), β-mercaptoethanol (0.0025%, all from Invitrogen) and bFGF (20 ng ml⁻¹, provided by Leibniz University Hannover, Institute of Technical Chemistry), or TeSR-E7 medium (Stem Cell Technologies). Single emerging iPSC clones were mechanically transferred onto fresh feeder layers in PSC medium between days 9 and 20 or onto Synthemax-coated plates on day 14 in TeSR-E8 (Stem Cell Technologies) and further expanded.

For immunocytology, cells were fixed with 4% paraformaldehdye and stained by standard protocols using antibodies specific to OCT4 (1:100, Santa Cruz Biotechologie), NANOG (1:500, Abcam), TRA-1-60 (1:100, Abcam), SSEA-3 and SSEA-4 (1:100 and 1:70, Iowa Hybridoma Bank). To test multi-lineage differentiation, cell were culture in DMEM plus 10% fetal bovine serum for up to 20 days followed by staining with antibodies specific to Troponin T (1:100, Thermo Scientific), Sarcomeric-α-Actinin (1:800, Sigma), α-Fetoprotein and Sox17 (1:300 and 1:200, both R&D Systems), Desmin (1:20, Progen Biotechnik), and β-3-Tubulin (1:400, Upstate). Appropriate secondary antibodies Cy2-labelled donkey anti-mouse IgG, Cy3-labelled donkey anti-mouse IgG or donkey anti-mouse IgM (1:200, Jackson Immunoresearch Laboratories) were applied followed by counterstaining with 4,6-diamidino-2-phenylindole and analysis with an Axio Observer A1 microscope (Carl Zeiss).

**hPSC differentiation.** Confluent, feeder-free hPSCs were detached using accutase and inoculated as single cells at $3.3 \times 10^5$ cells per ml in 1.5 ml mTeSR1 supplemented with Rho-Kinase inhibitor Y-27632 $(10 \, \mu\text{M})$[56] in suspension culture 12-well plates (Greiner) for aggregate-based differentiation, or at $2.66 \times 10^5$ cells per ml in 1.5 ml on geltrex-coated (Life Technologies) plates (Nunc) for monolayer-based differentiation. A volume of 1 ml mTeSR medium was refreshed on days −2 and −1. Differentiation of formed aggregates was initiated (day 0) by changing medium to RPMI 1640 supplemented with B27 without insulin (1%, Life Technologies). During the first 24 h of differentiation, CHIR was added as indicated at concentrations of 5–15 μM in a medium volume of 1–3 ml. Thereafter, cells were kept in 1.5 ml medium volume, unless otherwise noted. On day 3, IWP2 (5 μM, Tocris) was added for 48 h. Aggregates were kept in RPMI 1640 containing B27 from day 7–10.

Additional growth factors and small molecules were used at following concentrations: bFGF (10 ng ml⁻¹)[57]; Activin A (6 ng ml⁻¹); BMP2 (1–100 ng ml⁻¹, both Peprotech); dorsomorphin (0.5–2 μM); forskolin (10 μM); insulin (1–20 μg ml⁻¹); LEFTY1 (100–200 ng ml⁻¹, all Sigma); BMP4 (5 ng ml⁻¹); SFRP1 (0.1–1 μg ml⁻¹); BMPER (0.5–50 μg ml⁻¹, all R&D Systems); Cer1 (1–10 μg ml⁻¹, Erik Martinez-Hackert); IWR-1 (4 μM, Stemgent); and WAY-262611 (1 μM, Merck Millipore). CHIR99021 was kindly provided by the Institute of Organic Chemistry, Hannover.

**Medium enrichment (Amicon).** Differentiation was induced using CHIR as described above. Supernatant was collected after 6 and 24 h, and subsequently

enriched by centrifugation at 4,500g for 15–20 min using centrifugal filter tubes (Amicon). The enriched medium was stored at −20 °C or directly added to experimental conditions as indicated.

**Small interfering RNA.** Single-cell hPSCs were seeded on geltrex-coated plates at $4.5 \times 10^5$ cells per $cm^2$ in E8 medium supplemented with Y-27632 (10 μM), respective small interfering RNAs (siRNAs; 100 nM, Ambion Silencer Select pre-designed) and Lipofectamine 3000 (0.75%, Invitrogen). AlexaFluor546-labelled scrambled siRNA (Qiagen) and lipofectamine-only-treated cells were used as controls. After 24 h cells were washed with PBS and differentiation was performed as described above. The sequences for CER1 and LEFTY1 were 5′-ACCACU UCAUGUUCAGAAAtt-3′ and 5′-GCUCUGUGCUCUCUAGUGAtt-3′, respectively.

**Flow cytometry.** Cells were dissociated using collagenase B (1 mg ml⁻¹, Roche) for 5–45 min at 37 °C (ref. 2). For staining of cardiac markers, $1.5 \times 10^5$ cells were fixed and permeabilized according to the manufacturer's instructions (Fix & Perm; An der Grub). For other intracellular staining, cells were fixed and permeabilized by incubation in cold methanol (90%) for 15 min. Anti-cardiac Troponin T (1:200, clone 13-11, Thermo Scientific), anti-sarcomeric α-actinin (1:800, EA53, Sigma-Aldrich), anti-myosin heavy chain α (1:20, MF20, Hybridoma Bank), anti-T-brachyury (1:50, N-19, Santa Cruz), anti-vimentin, anti-CDX2, anti-TRA-1-60 (all 1:100, EPR3776, AMT28, TRA-1-60, respectively, Abcam), SSEA3, SSEA4 (1:100, MC-631 and MC-813-70, respectively, Hybridoma Bank) and respective isotype controls (DAKO) were detected using appropriate Cy3- or Cy5-conjugated antibodies (1:200; Jackson ImmunoResearch). EpCAM-FITC (1:100; EBA-1), NCAM-PE-CF594 (1:50; B159, both BD biosciences), CD90-APC (1:100, 5E10, Life Technologies), cKIT-PE (1:100; both eBioscience) and CXCR4-APC (1:100; both eBioscience) were incubated for 30 min at 4 °C. Data were acquired on an Accuri C6 flow cytometer (BD Biosciences) and analysed using FlowJo software (Treestar, Ashland, USA).

**Western blotting.** SDS–PAGE gels were blotted onto nitrocellulose membranes and incubated with the respective primary antibodies (anti-SFRP1, Cell Signaling Technologies, clone D5A7, 1:500; anti-DKK1, R&D Systems, AF1096, 1:2,000). As secondary antibodies fluorescently labelled IRDye800 and 680 (LI-COR, Lincoln, NE) were used. Fluorescence detection was performed using an Odyssey infrared imaging system (LI-COR) and blots were further analysed using the Odyssey v3.0 software (LI-COR).

**Protein array.** A customized human Quantibody protein array (QAH-CUST, Hölzel Diagnostika) covering 40 different factors was performed according to the manufacturer's instruction and scanned on an Axon 4000B. Data were extracted using GenePix and processed using Q-Analyzer v8.40.4.

**Microarray.** A customized whole-human genome oligo microarray 4 × 44 K v2 (design ID 054261, Agilent Technologies) covering roughly 26,000 human transcripts was hybridized with Cy3-labelled cRNA synthesized by the 'Quick Amp Labeling kit, One Color' #5190-0442, Agilent Technologies. cRNA fragmentation, hybridization and washing steps were carried out exactly as recommended in the 'One-Color Microarray-Based Gene Expression Analysis Protocol V5.7'. A unit of 1,650 ng of each labelled cRNA population were used for hybridization.

Slides were scanned on the Agilent Micro Array Scanner G2565CA (pixel resolution 3 μm, bit depth 20). Data extraction was performed with the 'Feature Extraction Software V10.7.3.1' using a modified version of the recommended default extraction protocol file 'GE1_107_Sep09.xml' in which the minimal number of replicates to calculate Population Outliers was set to 5. Data were analysed using Qlucore Omics Explorer 3.0 (Qlucore AB, Lund, Sweden) for principal component analysis and heatmap generation. The RCUTAS filter tool (Research Core Unit Transcriptomics of Hannover Medical School) was used for the identification of >2- and >5-fold regulated genes. Gene ontology analysis of >5-fold regulated genes was performed using webgestalt software (http://bioinfo.vanderbilt.edu/)[58]. The microarray data were deposited under accession number E-MTAB-5051 in the ArrayExpress database (www.ebi.ac.uk/arrayexpress).

**Ultra-performance liquid chromatography-based quantification of CHIR99021.** A volume of 5 μl of each sample were injected on a ultra-performance liquid chromatography (UPLC) column (35 °C column temperature, nuclear gravity, 50 × 2 mm, 1.8 μm, Macherey&Nagel, Germany) and analysed by UPLC–ESI–MS (Acquity UPLC coupled with Q-Tof premier electrospray ionization (ESI)–MS, software Masslynx 4.1, all components by Waters, UK) using a linear gradient of solvent A (double distilled water; 0.1% formic acid) and solvent B (Fluka methanol LC–MS CHROMASOLV; 0.1% formic acid): 0 min (95% A and 5% B); 2.5 min (5% A and 95% B); 6.5 min (5% A and 95% B); 6.6 min (95% A and 5% B); and 8 min (95% A and 5% B, end of run) at a flow rate of 400 μl min⁻¹. CHIR99021 was detected with a retention time of 1.69–1.99 min by ESI–MS (3 kV capillary voltage, 250 °C desolvation temperature and 650 l h⁻¹ nitrogen as desolvation gas) as positive-charged ion (M + H +). For quantification, the selected ion chromatogram for the mass range 464.5–465.5 Da (M + H +) was generated and the signal at the proposed retention time was integrated.

**LC–MS/MS and automated MS data analysis.** Secreted proteins were precipitated from cell culture supernatants applying a carrier-assisted tri-chloroacetic acid precipitation[59] and subsequently separated by SDS–PAGE. Each gel lane was cut into small pieces that were subjected to tryptic digestion according to standard procedures[60] (Supplementary Fig. 7). As described before[54] peptides were extracted, separated and analysed by reversed-phase chromatography using a nano-flow ultra-high-pressure liquid chromatography system (RSLC, Thermo Fisher Scientific) coupled online to an LTQ-Orbitrap Velos mass spectrometer (Thermo Fisher Scientific). For RSLC, peptides were trapped on a trapping column (3 μm C18 particle, 2 cm length, 75 μm inner diameter, Acclaim PepMap, Thermo Fisher Scientific) and separated using a reversed-phase separating column (2 μm C18 particle, 50 cm length, 75 μm inner diameter, Acclaim PepMap, Thermo Fisher Scientific). Peptide elution was performed with a multi-step binary gradient: linear gradient of buffer B (80% acetonitrile (ACN) and 0.1% formic acid) in buffer A (0.1% formic acid) from 4 to 25% in 115 min, 25 to 50% in 25 min, 50 to 90% in 5 min, 90% B for 10 min and switched to 4% B within 30 min for reconditioning of the column at a flow rate of 250 ml min⁻¹ and a column temperature of 45 °C. For MS, overview scans were acquired at a resolution of 60,000 ($m/z$ = 400) in the mass range of $m/z$ 300–1,600, and the 10 most intensive two- and threefold charged ions were fragmented by collision-induced fragmentation at an activation time of 10 ms and an activation Q of 0.250 in the LTQ. Fragment ion spectra were recorded in the LTQ at a normal scan rate and stored as centroid $m/z$ pairs. Raw data were processed with the MaxQuant proteomics software suit[61] version 1.5.0.22. Peak lists were searched against the human International uniprot protein sequence database (uniprot-homo + sapiens) downloaded on 09 July 2015 at www.uniprot.org at a false discovery rate of 1%. Proteins marked as 'only identified by site', 'reverse' or 'contaminant' were removed from MaxQuant output files. Quantitative comparison between different samples was performed based on label-free quantification intensities calculated by MaxQuant. The raw data were deposited to the ProteomeXchange Consortium via the PRIDE partner repository (www.ebi.ac.uk/pride) with the data set identifier PXD004874. The processed data are listed in Supplementary Data 1.

**β-catenin suspension bead array-based assay.** Cells were collected by centrifugation at 2,000 r.c.f. at 4 °C. Cell homogenization was performed under native conditions in a 10-fold volume of lysis buffer (150 mM NaCl, 50 mM Tris (pH 7.4), 1% Triton X-100, 1 × Complete (Roche), 1 × Phosphatase Inhibitor Cocktail I and II (SIGMA-Aldrich), and Benzonase (2.5 units per ml, Novagen)). Homogenates were incubated for 1 h on ice before insoluble cell fragments were separated by centrifugation (20 min, 4 °C, 13,000 r.c.f.). Supernatants were analysed by bead array-based sandwich immuoassays for total β-catenin, β-catenin phosphorylated at Ser33/ Ser37/Thr41, β-catenin phosphorylated at Ser45, β-catenin phosphorylated at Ser552 and β-catenin phosphorylated at Ser 675 as described earlier[32]. Free β-catenin was investigated by bead array-based glutathione S-transferase-pull-down assays using glutathione S-transferase fusion proteins for β-catenin and Tcf (ICAT), transcription factor 4 (TCF4) and the cytoplasmic domain of E-cadherin as baits[32,33].

**Statistics.** All data are presented as mean ± s.e.m. Unless otherwise noted, statistical significance was determined by one-way analysis of variance followed by Bonferroni's multiple comparison post-test. Statistical significance was assigned as indicated in the figure legends.

**Data availability.** The authors declare that all data supporting the findings of this study are available within the article and its Supplementary Information files or from the corresponding author on reasonable request. Microarray data have been deposited in the ArrayExpress database under accession number E-MTAB-5051. Raw data for LC–MS/MS have been deposited in ProteomeXchange Consortium via the PRIDE partner repository under accession code PXD004874.

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

## Acknowledgements

We thank T. Scheper for providing bFGF, D. Elliott for the hES-NKX2.5 line, A. Elefanty and E. Stanley for the hES-MIXL1 line, and J. Hoepfner and T Cantz for the H9-TCF line. We thank the RCU Transcriptomics of the Hannover Medical School for performing the microarrays, the Mass Spectrometry Laboratory at the Research Core Unit Proteomics, Hannover Medical School, headed by Andreas Pich for performing the mass spectrometric analyses and F. Stahl for providing the GenePix scanner. This work was funded by the DFG (Cluster of Excellence REBIRTH DFG EXC62/1 and grant no. ZW 64/4-1, MA 2331/16-1), the BMBF (grant no. 13N12606), StemBANCC (support from the Innovative Medicines Initiative joint undertaking under grant agreement no. 115439-2, resources of which are composed of financial contribution from the European Union (FP7/2007-2013) and EFPIA companies in kind contribution) and TECHNOBEAT (European Union H2020, GA no. 668724). Henning Kempf was supported by Hannover Medical School internal program (HiLF) and by Joachim Herz Stiftung.

## Author contributions

H.K., R.Z. and R.O. designed the study; H.K., A.F., A.H., E.B., D.R.-D., K.S. and R.O. performed the experiments; H.K., F.F.R.B., R.O., A.H., U.M. and R.Z. analysed and interpreted the data; F.F.R.B performed the western blots and LC–MS/MS analysis. O.P. and T.J. performed the β-catenin assay; G.D. performed UPLC–MS analysis and provided CHIR99021; G.G. performed the karyogramm analysis; E.M.-H. provided recombinant CER1; U.M. and A.H. gave conceptual advice; H.K., R.Z. and M.C. wrote the paper. All authors approved the manuscript.
