## [Peer Review File · Nature Communications]

Reviewer #1 (Remarks to the Author)

In this manuscript, Kempf and colleagues investigate a striking technical phenomenon centred around culture conditions promoting cardiomyocyte differentiation in human pluripotent stem cells. The authors mainly focus on two parameters, the concentration of a WNT signaling agonist (CHIR99021) and the culture volume. In a series of systematic and well-presented experiments it is shown that different combinations of both these parameters give rise to strikingly different differentiation fates. These different outcomes are finally correlated with the progression through a primitive streak-like stage toward the beginning of differentiation, in a convincing manner. Overall, the phenomena investigated by the authors are of interest beyond a technical perspective, since they also relate to the control of early human development. Several points, however, would require modification and improvement.

Major specific points

- The rationale for analyzing the "secretomes" of differentiating cells has not really become clear to me and the data do not seem to be too revealing, as also admitted by the authors - "complex" - p10, "question [...] still remains" - p12. For example, the starting idea of this analysis was to use conditioned medium to rescue cardiac differentiation in the high-WNT / high-volume condition ("15/3"). However, the secreted factors identified were later not used for this purpose but rather, the authors performed knockdown experiments to rescue yet a different culture condition ("7.5/1"). It would generally be easier to appreciate any impact of a secreted factor by supplementing it to the medium rather than silencing it using genetic manipulation. Moreover, the factors identified (Lefties, Cerberus) are actually factors strongly expressed and secreted by undifferentiated hPSCs - not differentiated ones. This is also reflected by the fact that the authors used a very early time-point of analysis - 6 hours of differentiation. At this early time-point, the hPSCs are not yet differentiated and will still express the above ligands at high levels. So, the results are based on an undifferentiated hPSC profile and not on differentiating cells. It is therefore misleading to put those factors forward as "anteriorizing" patterning molecules supposedly released by differentiating hPSCs. At least, then the authors would have to perform the key experiment predicted by their model (Fig. 6b), namely, to restrict primitive streak progression under the 15/3 condition by supplementing Lefties and Cer, and rescue the cardiac differentiation defect.

Otherwise, I believe the answer is not in the secretome of the cells. Rather, the authors should consider the following possibility: That the key determinant for cardiac differentiation in their system is s.th. like the number of CHIR molecules (not CHIR concentration) per input cell number. In this view, the number of CHIR molecules are similar when lowering the culture volume but concomitantly increasing the CHIR concentration, such that the differentiation outcomes will also be similar (green color-coded conditions in this manuscript). In the 7.5/1 condition, however, the number of CHIR molecules per cell is reduced and hence, cardiac differentiation fails. Likewise, in the 15/3 condition, the number of CHIR molecules per cell is overdosed, which explains the reduced yield of cardiomyocytes.

The authors themselves already performed an elegant experiment which argues in favor of such a view (Fig. 1e, condition b): The third condition in this experiment appears to be compatible with cardiomyocyte differentiation because both cell numbers and volume (i.e. absolute number of CHIR molecules in that latter case) are increased at the same time, as compared to condition 2... In this view, therefore, everything is about the overall WNT dose that the cells receive at the beginning of differentiation, as also partially reflected in the authors' model - except that the culture volume (or cell density) should not be considered a relevant parameter by itself: The effective WNT dose acting on the cells would overall be determined by (i) the number of CHIR molecules, (ii) the number of input cells, and (iii) the time of exposure (as supported by data in Fig. 5). This is further supported by the positive effect of IWR on the 15/3 condition in Fig. 4c. I would strongly recommend restructuring this promising work according to these three parameters, at the expense of the secretome data.

- Along these lines, can the authors be certain that the 15/1 and 15/3 conditions do not differ in

their WNT response? The data of Fig. 3 are surprising in this regard. Can the authors rule out that these assays may be saturated? Furthermore, what was the time-point of analysis in these experiments? Also, the corresponding paragraph on p10 does not read too well. I think the gene expression profiles that the authors present as well as the positive effect of IWR does strongly argue for a WNT overdose scenario under the 15/3 condition. It would be helpful if the authors could revisit this point by using different time-points of analysis or alternative WNT reporter assays. This would further strengthen their primitive streak progression model (non-cardiac cell fate at 15/3 due to too high WNT dose per cell).

Other points

- The writing style should be improved. Title and abstract are not well-written. (Can a "bulk cell density" be considered a "method"? / "by demonstrating BCDs' decisive role...".) Overall, this works in part reads too technical, i.e. more biological meaning would be appreciated. Moreover, the authors should try to make an effort to better guide the reader through this work - without forcing him/her to read figure legends through in detail, for instance. To pick just one example: What is WAY-262611 (p9) and how does it act?
- The immunofluorescence stains of Fig. S3 are all not clear. They should be shown at significant higher magnification / cropped images, such that the intracellular staining pattern becomes evident.
- Fig. S5c: How am I supposed to interpret this blot? I cannot appreciate any positive bands here...
- Several figures are mislabeled in text (e.g. Fig. S1f, S3a). Moreover, some figure panels are not mentioned in the main text or mentioned earlier than preceding panels of the same figure.

Reviewer #2 (Remarks to the Author)

This article examines the interplay between cell density and WNT pathway inhibition with Chir on mesoendoderm specification, gene expression, secretome and ultimately cardiomyocyte formation using genetically engineered hPSC reporter lines. The effect of the secretome is also examined by varying the culture volume from 1 to 3 ml. This points to a model wherein Chir and BMP act in an antagonist manner with cell density (+ LEFTY1 + CER1). This is supported by siRNA knockdown experiments against LEFTY1 and CER1, as well as small inhibition of DKK1.

The work is generally well performed and is shown to be reproducible across multiple hPSC lines. The difficulty I have with the study, however, is that it lacks novelty in what is a crowded field. The influence of cell density of lineage specification and cardiomyocyte formation is well documented from many earlier studies using EBs and monolayers (e.g. Ng et al., Blood 2005; BurrIDGE et al., Stem Cells 2007; BurrIDGE et al., PLoS One 2011; BurrIDGE et al., Nature Methods 2014). Modulation of signalling by BMP, dorsomorphin and DKK1 to induce mesoderm and cardiac differentiation are well described by numerous reports including Yang et al., Nature, while the interplay between LEFTY1, CER and BMP is well described, particularly in the mouse embryo. While omics approaches and a model mechanism are used to 'beef up' the manuscript, the data produced really only serves to confirm what is already known.

Reviewer #3 (Remarks to the Author)

The authors describe a novel patterning methodology to utilize bulk cell density (BCD) as a key parameter to influence stem cell differentiation. The authors go on to conclude that BCD is an underappreciated (and underreported) factor in many publications on hPSC differentiation, and provide recommendations and methodologies for both reporting BCD within the culture format and also minimizing the effect of BCD by using specific culture format (monolayer expansion/differentiation platform). The authors provide a compelling story and a trove of data including flow cytometry, imaging, western blotting, protein arrays, mRNA microarray, and mass spectrometry to support their claims. Overall the study seems well designed, but given the expansive amount of data, there are experimental details lacking in the description of the study in

some areas.

Specifically, I was asked to review the MS data collection methodologies for completeness and transparency. The description of the measurement of CHIR99021 by mass spectrometry appears to have sufficient completeness and transparency. The below is a more complete review of the proteomics section. Much of the expanded details will likely need to be incorporated into supplemental information.

Insufficient details are provided in the Methods for the mass spectrometry and MS data analysis section for the cell culture supernatant. Cell culture supernatants are notoriously complex samples with wide dynamic range, therefore a gel image characteristic of the samples analyzed prior to band isolation would be a helpful reference. There are no details provided on the LC-MS/MS data collection parameters, such as chromatography setup (column, mobile phase, gradient, flow rate, temperature) or mass spectrometer settings (data acquisition resolution, speed, AGC settings, etc). Much of this could be clarified by providing the raw data available for download at one of several MS data repositories, such as MassIVE (<https://massive.ucsd.edu/ProteoSAFe/static/massive.jsp>). Additionally, the .fasta protein sequence database the authors used (ipi.HUMAN.v3.68) for database protein searching was last updated sometime prior to 2010, and is no longer available for download publicly. This will make it difficult for anyone to reproduce the author's results even if they did have the raw data, and it is curious why the authors would choose to use a database and genome/proteome annotation that is more than six years old. By contrast, using the curated uniprot (www.uniprot.org) or NCBI curated database (<https://www.ncbi.nlm.nih.gov/protein>) would allow much more transparency. It is not clear whether the investigators searched against a database of common contaminants which could reduce false positive discoveries. Regardless of the database used, the authors should be required to provide a list of peptide (along with protein) identifications, including the monoisotopic m/z, retention time, any modifications on the peptide, and the database identification score. This is a minimum in order to assess results quality.

Reviewers' comments:

Reviewer #1 (Remarks to the Author):

In this manuscript, Kempf and colleagues investigate a striking technical phenomenon centred around culture conditions promoting cardiomyocyte differentiation in human pluripotent stem cells. The authors mainly focus on two parameters, the concentration of a WNT signaling agonist (CHIR99021) and the culture volume. In a series of systematic and well-presented experiments it is shown that different combinations of both these parameters give rise to strikingly different differentiation fates. These different outcomes are finally correlated with the progression through a primitive streak-like stage toward the beginning of differentiation, in a convincing manner. Overall, the phenomena investigated by the authors are of interest beyond a technical perspective, since they also relate to the control of early human development. Several points, however, would require modification and improvement.

Major specific points

- The rationale for analyzing the "secretomes" of differentiating cells has not really become clear to me and the data do not seem to be too revealing, as also admitted by the authors - "complex" - p10, "question [...] still remains" - p12. For example, the starting idea of this analysis was to use conditioned medium to rescue cardiac differentiation in the high-WNT / high-volume condition ("15/3"). However, the secreted factors identified were later not used for this purpose but rather, the authors performed knockdown experiments to rescue yet a different culture condition ("7.5/1"). It would generally be easier to appreciate any impact of a secreted factor by supplementing it to the medium rather than silencing it using genetic manipulation. Moreover, the factors identified (Lefties, Cerberus) are actually factors strongly expressed and secreted by undifferentiated hPSCs - not differentiated ones. This is also reflected by the fact that the authors used a very early time-point of analysis - 6 hours of differentiation. At this early time-point, the hPSCs are not yet differentiated and will still express the above ligands at high levels. So, the results are based on an undifferentiated hPSC profile and not on differentiating cells. It is therefore misleading to put those factors forward as "anteriorizing" patterning molecules supposedly released by differentiating hPSCs. At least, then the authors would have to perform the key experiment predicted by their model (Fig. 6b), namely, to restrict primitive streak progression under the 15/3 condition by supplementing Lefties and Cer, and rescue the cardiac differentiation defect.

Otherwise, I believe the answer is not in the secretome of the cells. Rather, the authors should consider the following possibility: That the key determinant for cardiac differentiation in their system is s.th. like the number of CHIR molecules (not CHIR concentration) per input cell number. In this view, the number of CHIR molecules are similar when lowering the culture volume but concomitantly increasing the CHIR concentration, such that the differentiation outcomes will also be similar (green color-coded conditions in this manuscript). In the 7.5/1 condition, however, the number of CHIR molecules per cell is reduced and hence, cardiac differentiation fails. Likewise, in the 15/3 condition, the number of CHIR molecules per cell is overdosed, which explains the reduced yield of cardiomyocytes.

The authors themselves already performed an elegant experiment which argues in favor of such a view (Fig. 1e, condition b): The third condition in this experiment appears to be compatible with cardiomyocyte differentiation because both cell numbers and volume (i.e. absolute number of CHIR molecules in that latter case) are increased at the same time, as compared to condition 2... In this view, therefore, everything is about the overall WNT dose that the cells receive at the beginning of differentiation, as also partially reflected in the authors' model - except that the culture volume (or cell density) should not be considered a relevant parameter by itself: The effective WNT dose acting on the cells would overall be determined by (i) the number of CHIR molecules, (ii) the number of input cells, and (iii) the time of exposure (as supported by data in Fig. 5). This is further supported by the positive effect of IWR on the 15/3 condition in Fig. 4c. I would strongly recommend restructuring this promising work according to these three parameters, at the expense of the secretome data.

Response: We highly appreciate reviewers overall view on our data and have thoroughly responded to all issues raised in the above section as follows.

Reviewer stated: "Otherwise, I believe the answer is not in the secretome of the cells. Rather, the authors should consider the following possibility: That the key determinant for cardiac differentiation in their system is s.th. like the number of CHIR molecules (not CHIR concentration) per input cell number."

We are thankful to the reviewer for highlighting this alternative hypothesis to explain our findings. In this rebuttal letter we would like to respectfully term the possibility that the number of CHIR molecules per input cell number determines the cell fate the 'chemical hypothesis'.

On the other hand, we will term the assumption that the bulk cell density (BCD)-dependent secretomes dictates PS-patterning the 'biological hypothesis'.

When we have observed the first results of the present study, we have intensively discussed the 'chemical hypothesis' as a potent explanation for our data. However, investigating this possibility by suitable experiments did not support this idea as follows.

If the 'number of CHIR molecules per cell' would be 'deterministic', one would assume that CHIR is depleted from the medium by a biochemical process over time; consequently a drop in CHIR concentration is expected. However, mass spectrometry (MS) analysis of the supernatant in the absence or presence of cells revealed an absolute constant CHIR99021 concentration even for up to 48h as shown in Supplementary Fig. 4a. This clearly reveals stability and maintenance of CHIR99021 at the

respective concentration in the medium at experimental conditions, thereby verifying unrestricted exposure of every cell in culture to a given CHIR99021 concentration at any time irrespective of the BCD.

In other words, our data suggest that a surplus number of CHIR99021 molecules is available to all cells throughout the 24h exposure in our experiments, irrespective of the input cell numbers / medium volume. This, in our opinion, excludes a mechanism where the number of CHIR molecules per cell dictates the differentiation outcome.

It is worth emphasizing that other data in the manuscript, in our view, also contradict the chemical hypothesis. As shown in Fig. 4a/4b and 5a, interference in the secretomes rescued the cardiac phenotype at 7.5/1 and 15/3 conditions, respectively. Importantly, in these experimental interventions, the CHIR concentration as well as the number of CHIR molecules per cell remains unchanged during the entire 24h incubation period; the chemical hypothesis would thus predict an unchanged differentiation outcome, which clearly is in contrast to our findings (Fig. 4a/4b and 5a).

On the other hand, the biological hypothesis suggests a “reset” of paracrine factors must impact on cell differentiation, which tallies with our results. This view is further supported by our “reversal experiments”, wherein the enriched proteome (Fig. 5a) or defined factors including LEFTY1 and/or CER are supplemented to respective cornerstone conditions (Fig. 5f and Supplementary Fig. 5j). In these experiments “the input number of CHIR molecules per cell” is unaltered compared to controls but the paracrine environment, only is modulated, which has substantial consequences for directing hPSC priming (Fig. 5f).

However, inspired by reviewers’ precise questions on the “CHIR dose” effect, we have performed additional experiments to unequivocally distinguish between the “chemical” (i.e. CHIR dose-dependent) versus “biological” (secretome-determined) hypothesis. We propose that a chemical mechanism will evoke a response (in a relevant assay) in a cell type-independent manner. In contrast, a biological / paracrine mechanism should be rather cell type-dependent. We thus analyzed the WNT pathway activity in alternative cell types but using a uniform reporter readout. HEK293 cells were transfected with a TCF/LEF WNT reporter construct - following CHIR treatment at different concentrations and applying respective medium volume combinations (BCD conditions) equivalent to our experimental setup in the manuscript. As shown in Rebuttal Figure 1 (left panel) below, we did not observe any CHIR dose-dependent effect in the HEK cell reporter assay although the cells were clearly responsive to CHIR stimulation, suggesting proper WNT pathway activity.

In contrast, using the human ES cell line H9 carrying the same TCF/LEF WNT promoter/reporter system (Hoepfner et al., 2016) we have observed an unequivocal response of this reporter assay to culture volume modulations (Rebuttal Figure 1, right panel), closely reflecting the data on pluripotent cells in our manuscript.

Rebuttal Fig. 1: HEK293T cell were transfected with a 7TCF reporter construct and subjected to 2.5-15μM CHIR treatment at different volumes equivalent to experimental settings of the cardiac differentiation protocol (left). The hPSC reporter line H9 (Hoepfner et al., 2016) was subjected to 7.5μM CHIR treatment in 1 and 3 ml medium equivalent to experimental settings of the cardiac differentiation protocol (right).

These cell-type dependent results - in our view - are in clear disagreement with the chemical hypothesis / CHIR molecule response. In contrast, the data strongly suggests a key role of the BCD-dependent accumulation of paracrine factors, which in HEK293 cells showed no impact on WNT pathway activity whereas pluripotent H9 cells clearly reflect our data in the paper.

Taken together, we are convinced that our results strongly support the assumption that the BCD-dependent, differential “secretomes” dictate the observed differentiation outcome in our study. We therefore respectfully request to keep the secretome analysis in the manuscript as it constitutes an integral part of the study and is key to interpret our findings.

- In other major specific points above reviewer stated:

Reviewer stated: "Moreover, the factors identified (Lefties, Cerberus) are actually factors strongly expressed and secreted by undifferentiated hPSCs - not differentiated ones. This is also reflected by the fact that the authors used a very early time-point of analysis - 6 hours of differentiation. At this early time-point, the hPSCs are not yet differentiated and will still express the above ligands at high levels. So, the results are based on an undifferentiated hPSC profile and not on differentiating cells. It is therefore misleading to put those factors forward as "anteriorizing" patterning molecules supposedly released by differentiating hPSCs. At least, then the authors would have to perform the key experiment predicted by their model (Fig. 6b), namely, to restrict primitive streak progression under the 15/3 condition by supplementing Lefties and Cer, and rescue the cardiac differentiation defect."

Response: We strongly agree with reviewer's overall notion that our results are significantly influence by the undifferentiated hPSC secretion profile. We have stated this in the manuscript as follows:

"Recently, single-cell RNA-sequencing of the human blastocyst revealed that LEFTY1, together with other key components of the TGF β signaling, is enriched in the human epiblast (Blakeley et al., 2015). TGF β signaling was found to control NANOG expression in epiblast cells revealing pathways' requirement in pluripotency, whereby conservation of pluripotency control between the human embryo and hESC was highlighted (Blakeley et al., 2015)."

However, following reviewers vested request, we have carried out additional experiments to substantiate our observation that specific paracrine factors in particular LEFTY1 and CER1 trigger PS-patterning at earliest stages of differentiation. Besides showing the "cardiac rescue" of the 7.5/1 condition by siRNA-mediated knock-down of CER1 and LEFTY1, we now also show differential response of MIXL1 after 24h of differentiation in consequence to these knock-down experiments (novel data presented in Figure 5e). These additional data support the view that CER1 and LEFTY1 indeed have a function in restricting primitive streak-like progression in hPSC. Conversely, we have now also added human recombinant CER1 and LEFTY1 to the 15/3 conditions as requested by the reviewer. These novel data, which are shown in Fig. 5f, confirm the specific role of these factors in regulating primitive streak patterning and progression.

We also provide new data in Supplementary Fig. 5j showing, by endpoint analysis, that supplementation of the individual factors, at least partially, rescue the cardiac differentiation outcome on day 10 i.e. ~75% for CER1 and ~30% for LEFTY1 of 15/3 conditions as compared to the level of cardiomyocyte induction in the 15/1 cornerstone.

The reviewer has also highlighted that LEFTY and CER1 are factors that are readily expressed in hPSCs rather than being upregulated during differentiation. Although CER1 due to our analysis is only upregulated after induction of differentiation (Supplementary Fig. 5a, Fig5c), we fully share the view that the expression of LEFTY1 at the pluripotent state is a key feature that impacts on paracrine control of PS-patterning and progression. Due to this fact, differential BCD-dependent accumulation of LEFTY1 occurs swiftly thereby immediately modulating (i.e. restricting) CHIR-triggered progression of differentiation.

However, regarding the known and putative functions of CER1 and LEFTY in human pluripotent cells we would also like to make the following notions:

The relative high expression of LEFTY1 in hPSCs has been interpreted as a consequence (and a marker) of their epiblast-like state in vitro (Blakeley et al., 2015; O'Leary et al., 2012). In contrast, no expression in the inner cell mass (ICM) of the mouse blastocyst stage embryo is detected (Blakeley et al., 2015; O'Leary et al., 2012; Takehara et al., 2012). Similarly, Cerberus is not a "classical" gene that is expressed by human pluripotent stem cells but is rather associated with an early differentiation state (Joo et al., 2014), is an established marker for definitive endoderm (Funa et al., 2015; Iwashita et al., 2013; Morrison et al., 2008). In addition, CER1 and LEFTY1 are expressed in the visceral endoderm of the mouse embryo controlling patterning in the anterior region of the embryo (Bertocchini and Stern, 2002; Perea-Gomez et al., 2002). In line with this, spatially-resolved transcriptome analysis of early mouse embryo (Peng et al., 2016) showed highest level of CER1 and LEFTY1 in the most anterior region of the primitive streak (see Rebuttal Figure 2). Interestingly, CER1 and LEFTY1 were identified as key candidates contributing to the heterogeneity in hPSC cultures (Hough et al., 2014).

Taken together, neither CER1 and nor LEFTY1 are typical pluripotency markers, but are expressed in the transition state from epiblast towards primitive streak and particularly control the anterior patterning by restricting primitive streak formation in this process (Perea-Gomez et al., 2002), which overall fits very well to our findings.

Rebuttal Figure 2: Spatial expression pattern of the mouse embryo at late mid-streak stage showing highest LEFTY1 and CER1 in the most anterior region of the primitive streak (red dot). Legend: anterior (A), posterior (P) and left lateral (L) and right lateral (R) (Peng et al., 2016).

<http://www.picb.ac.cn/hanlab/itranscriptome/PatternSearchByGene/>

Reviewers comment: *Along these lines, can the authors be certain that the 15/1 and 15/3 conditions do not differ in their WNT response? The data of Fig. 3 are surprising in this regard. Can the authors rule out that these assays may be saturated? Furthermore, what was the time-point of analysis in these experiments? Also, the corresponding paragraph on p10 does not read too well. I think the gene expression profiles that the authors present as well as the positive effect of IWR does strongly argue for a WNT overdose scenario under the 15/3 condition. It would be helpful if the authors could revisit this point by using different time-points of analysis or alternative WNT reporter assays. This would further strengthen their primitive streak progression model (non-cardiac cell fate at 15/3 due to too high WNT dose per cell).*

Response: We highly appreciate reviewers' critical view on the WNT response at 15 μ M CHIR conditions and the specific request for revising the respective dataset. The β -catenin activity assessment shown in Fig.3, which represents endpoint analysis after 24h of CHIR treatment, highlights the relative high WNT response in our experimental system i.e in hPSCs. The striking lack of differential β -catenin activity between 15/1 and 15/3 conditions might indeed indicate assay saturation.

However, with respect to this issue, we would like to focus reviewer's attention to the "time-course analysis of β -catenin signaling activity and phosphorylation during the first 24h of differentiation", which is presented in Supplementary Fig.4b. This assessment clearly shows that the WNT response at 15/1 and 15/3 are almost entirely overlapping throughout the course of analysis, including early time-points at which saturation of the assay signal can be excluded. Please see, for example, the data for TCF and ECT assessment at 6h (360 min) in Supplementary Fig 4b: The 15/1 and 15/3 data-points are overlaid at this time-point (as well any other time analyzed) whereas the signals are readily distinct from 7.5/1 and 7.5/3 conditions, respectively.

Furthermore, it is worth highlighting that we have included six distinct indicators of WNT pathway activity in our assay (fishing of TCF, ECT, ICAT as well as S675 and S552 phosphorylation) of which each individual parameter suggests similar WNT pathway activity at 15 μ M CHIR independent of the medium volume. Given our extensive experience with these assays, we can essentially exclude that all measurements were technically saturated for each individual parameter, particularly as the overall signal intensity covers a broad range (MFI for S552 <2000 and >6000 for TCF). Together, this unequivocally shows that the similarity between 15/1 and 15/3 with respect to WNT response is no result of any technical saturation limits of our assays.

In consequence, these data support our view that 15 μ M CHIR supplementation induces high WNT stimulation, which – at least with respect to the β -catenin activity shown in Fig. 3 and Suppl. Fig.4b - is insensitive to the differential paracrine conditions at 15/1 and 15/3.

This further suggests that the differential biological response between 15/1 and 15/3 - which we have observed by numerous assays at any time-point along the differentiation process - is largely mediated by WNT-independent mechanisms. We like highlighting to the reviewer that this assumption is not contradicted with our data on the co-supplementation of the WNT inhibitor IWR1 in parallel to CHIR treatment (Fig. 4h). IWR1 is a chemical modulator of the stability of AXIN thereby directly impacting on the β -catenin stability and activity, irrespective of paracrine WNT pathway modulation (Chen et al., 2009). In line with this, supplementation of the chemical inhibitor IWP2 – which is known to block the secretion of WNTs by inhibition of PORCUPINE (Chen et al., 2009) - resulted in a negligible biological response in our cornerstone conditions (novel Supplementary Figure 5i). These observations tallies with the view that secreted WNTs do not play a pivotal role in CHIR-induced PS-priming in our model. This is further supported by the ineffective supplementation of the secreted WNT pathway antagonist DKK1 (Fig 4c, Supplementary Fig 5d) despite the massive cornerstone-specific differences in expression and accumulation of DKK (Supplementary Fig 5a-d); supplementation of another secreted WNT antagonist, SFRP1, showed minor effects in our system as well (Fig 4c).

This is now more explicitly stated on page 10 of the revised manuscript as follows:

"BMP4 addition hardly perturbed the WNT pathway at the level of β -catenin activity (Supplementary Fig. 6)., However, some reduction of β -catenin activity was observed at 15/1 conditions suggesting a direct crosstalk between the BMP and WNT pathways as previously published (Itasaki and Hoppler, 2010) Opposing BMP signaling by dorsomorphin (DM)(Yu et al., 2008) resulted in significant NKX2.5-GFP reduction at 7.5/3 (Fig. 4g) suggesting cornerstone conditions' dependency on endogenous BMP signaling and subsequent anteriorization towards 7.5/1-like conditions. Notably, DM marginally affected 15/1 (Fig. 4g) in contrast to the significant impact of BMP4 supplementation outlined above (Fig. 4e).

BMP and WNT signaling (conducted through SMAD 1/5/8 and β -catenin, respectively) converge in controlling mesendoderm specification on the transcriptional level (Funa et al., 2015; Mendjan et al., 2014). Since antagonizing BMP by DM hardly affected the 15/1 and 15/3 conditions, we have postulated their predominant WNT dependence. Indeed, direct inhibition of WNT pathway activity by IWR1 (i.e. via stabilization of AXIN) increased NKX2.5-GFP+ levels at 15/3 (Fig. 4h) suggesting expected cornerstone anteriorization towards PCM priming (i.e. adjustment towards 7.5/3-like conditions). Accordingly, IWR1 disrupted cardiomyogenesis at 15/1, supposedly by anteriorization towards 7.5/1-like conditions. Notably, indirect inhibition of WNT pathway activity using IWP2 (a chemical inhibitor of PORCUPINE, which acts by disrupting secretion of WNTs) did not alter the cardiac differentiation outcome (Supplementary Fig. 5i) suggesting a minor role of secreted WNTs in specifying these cornerstone conditions."

Taken together, our data in Supplementary Fig.4 support the validity of our previous results showing that the 15/1 and 15/3 conditions do not differ in their WNT response, suggesting that their distinct biological response depends on the differential activity of other signaling pathways, in particular those regulated by secreted BMPs, LEFTYs and CER.

Reviewer stated:

- The writing style should be improved. Title and abstract are not well-written. (Can a "bulk cell density" be considered a "method"? / "by demonstrating BCDs' decisive role...".) Overall, this works in part reads too technical, i.e. more biological meaning would be appreciated. Moreover, the authors should try to make an effort to better guide the reader through this work - without forcing him/her to read figure legends through in detail, for instance. To pick just one example: What is WAY-262611 (p9) and how does it act?

Response: We thank the reviewer for these helpful and constructive suggestions.

We have subsequently adapted the title as follows: "Modulation of the bulk cell density as decisive parameter and simple method for mesendodermal patterning of human pluripotent stem cells".

Within the allowed word limit, we have also revised the entire manuscript to make it easier to read in particular without the necessity to study figure legends in detail.

Respective amendments include for example:

Re-formulation of a paragraph on 'β-catenin activity is BCD-sensitive at 7.5 μM but not at 15 μM CHIR' on page 8:

"Following confirmation of CHIR stability over 48h under experimental conditions (Supplementary Fig. 4a), multiplex arrays were applied for detailed status analysis of the WNT pathway effector β-catenin (Luckert et al, 2011; Luckert et al, 2012). A first response was readily detected after 2h (120 min) post CHIR exposure (Supplementary Fig. 4b), preceding endpoint patterns observed at 24h (1440 min; Supplementary Fig. 4b). Continuous increase in signal activity interestingly suggests diminished negative feedback of the WNT pathway in the presence of CHIR (Supplementary Fig. 4b) confirming other cell systems^{30, 31}.

Statistical analysis revealed minor impact of the BCD on the β-catenin status at 15 μM CHIR (Fig. 3 a-h) suggesting that differential patterning of 15/1 vs. 15/3 is largely independent of WNT pathway activity. In contrast, comparison of 7.5/1 and 7.5/3 revealed significantly higher β-catenin activity at 7.5/3 (low BCD) (Fig. 3). This suggests that canonical WNT pathway activity is more sensitive to BCD-dependent feedback at lower CHIR concentrations."

Re-formulation of a paragraph on 'Duration of CHIR treatment controls PS progression' on page 11:

"To investigate the time-dependence in this process, CHIR/BCD treatment was prolonged from 24h to 48h at 7.5/1. Significant NKX2.5-GFP⁺ increase from 7.4±2.0% to 56.8±3.5% (Fig. 5g) suggests PS progression by prolonged incubation, thereby apparently shifting DE-specification of 7.5/1 after 24h towards cardiomyogenesis after 48h."

Comment on the specificity of WAY-262611 on Page 9 which now reads:

"In contrast, the chemical DKK1 inhibitor WAY-262611 (Pelletier et al., 2009) significantly reduced NKX2.5-GFP⁺ levels (Supplementary Fig. 5e,f). Assuming specificity of WAY-262611, these findings might suggest that endogenous DKK1, per se, is required for proper mesendoderm formation/specification in hPSCs equivalent to early embryogenesis (Caneparo et al., 2007) but supplementation of recombinant DKK was ineffective in modulation differentiation in our system (Fig 4c, Supplementary Fig. 5d)."

Re-formulation of paragraph on 'Duration of CHIR treatment controls PS progression' on page 9:

"To investigate the time-dependence in this process, CHIR/BCD treatment was prolonged from 24h to 48h at 7.5/1. Significant NKX2.5-GFP⁺ increase from 7.4±2.0% to 56.8±3.5% (Fig. 5g) suggests PS progression by prolonged incubation, thereby apparently shifting DE-specification of 7.5/1 after 24h towards cardiomyogenesis after 48h. Conversely, reducing CHIR/BCD incubation from 24h to 18h at 15/3 increased NKX2.5-GFP⁺ from 46.3±2.3% to 71.08±4.7% (Fig. 5h)."

All amendments in the manuscript are highlighted in the Word Track modus.

Reviewer stated: *The immunofluorescence stains of Fig. S3 are all not clear. They should be shown at significant higher magnification / cropped images, such that the intracellular staining pattern becomes evident.*

Response: We fully agree with reviewer's request and have responded by increasing the image size and by including a higher magnification for each merged image of respective cell colonies and differentiated cells as well, as suggested. Subsequently, the single channel images were removed to improve clarity and structure of the figure. We hope that this matches reviewers' expectations.

Reviewer stated: *Fig. S5c: How am I supposed to interpret this blot? I cannot appreciate any positive bands here...*

Response: Indeed DKK1 shows no distinct band in the western blots. However, we would like to emphasize that the 15μM/3mL condition shows a respective signal around the expected size of 37kDa and a less intense signal at 7.5μM/3mL, respectively. In our extensive experience, the rather diffuse appearance of the DKK signal suggests pronounced protein glycosylation. Our observations are in line with published data showing that the calculated molecular weight of DKK1 is ~29kDa but the actual Dkk1 "band" appears at 35-45kDa on SDS-Page analysis (Niehrs, 2006); see also company links displaying specificity of respective DKK1 antibodies and the expected signal appearance on SDS gels: 1,2,3 (please STRG+click on respective number to open the link).

This explanation on the DKK signal in the western blot has been added to the description of Supplementary Fig 5c as follows:

'Representative Western blot for DKK1 of the 4 cornerstone conditions and controls from harvested supernatant after 24h confirms the data obtained in the microarray and proteinarray. The calculated molecular weight of DKK1 is

~29kDa, but the actual DKK1 band appears at 35-45kD suggesting pronounced glycosylation in line with previous work (Niehrs, 2006).'

Please further note that the well-defined BSA band at the top of the blot in Fig.S5c (which serves as a loading control) represents a useful internal standard enabling to state that the overall DKK1 signal (in terms of its width and intensity) indeed represents sample-specific levels of this factor.

The obtained pattern and signal strength of DKK1 on our SDS gels / western blots notably matches the independent assessment via the protein array, the transcriptome data and the MS data in Fig. S5a,b and Fig.5B. Together this strongly suggests an increased release of DKK1 at higher medium volume conditions (i.e. at lower BCD). We therefore propose keeping the western blots in the manuscript to complement these data.

Reviewer stated: *Several figures are mislabeled in text (e.g. Fig. S1f, S3a). Moreover, some figure panels are not mentioned in the main text or mentioned earlier than preceding panels of the same figure.*

Response: We are thankful to the reviewer for highlighting these issues. The labeling was adjusted accordingly.

Reviewer #2 (Remarks to the Author):

This article examines the interplay between cell density and WNT pathway inhibition with Chir on mesoendoderm specification, gene expression, secretome and ultimately cardiomyocyte formation using genetically engineered hPSC reporter lines. The effect of the secretome is also examined by varying the culture volume from 1 to 3 ml. This points to a model wherein Chir and BMP act in an antagonist manner with cell density (+ LEFTY1 + CER1). This is supported by siRNA knockdown experiments against LEFTY1 and CER1, as well as small inhibition of DKK1.

The work is generally well performed and is shown to be reproducible across multiple hPSC lines. The difficulty I have with the study, however, is that it lacks novelty in what is a crowded field. The influence of cell density of lineage specification and cardiomyocyte formation is well documented from many earlier studies using EBs and monolayers (e.g. Ng et al., Blood 2005; BurrIDGE et al., Stem Cells 2007; BurrIDGE et al., PLoS One 2011; BurrIDGE et al., Nature Methods 2014). Modulation of signalling by BMP, dorsomorphin and DKK1 to induce mesoderm and cardiac differentiation are well described by numerous reports including Yang et al., Nature, while the interplay between LEFTY1, CER and BMP is well described, particularly in the mouse embryo. While omics approaches and a model mechanism are used to 'beef up' the manuscript, the data produced really only serves to confirm what is already known.

Response: We fully respect reviewer's comment on our manuscript and have carefully reviewed the cited publications regarding the influence of the cell density on lineage specification and cardiomyocyte formation. We agree that the impact of the cell density per se has been noted. However, we would like to respectfully emphasize, that none of the previous studies systematically elucidated the effect on a molecular level and have developed a mechanistic understanding on how the cell density effect is mediated, particularly regarding its early impact on PS patterning. We would also like to stress, that we have addressed the cell density aspect in a much more specific and quantitative manner compared to previous work, using highly defined cell numbers, timing, and systematic comparison of different cell culture platforms. We thereby exclude the potential interference of cell-cell-interaction-dependent effects, but clearly reveal the deterministic role of the differential secretomes which, to our best knowledge, was not specifically addressed in any previous study.

We do agree, that the role of LEFTY1 and CER1 is generally described for the mouse embryo (Perea-Gomez et al., 2002) and mouse pluripotent stem cells (Kim et al., 2014). However, their role during human development and their decisive impact during the in vitro differentiation of hPSC is poorly understood. Furthermore, we would like to emphasize that hardly any study known to us addresses mechanisms underlying PS-like patterning of hPSC in vitro, in particular by such 'simple' but, as we show, highly deterministic means as modulating the medium volume / BCD. Given the thorough revision and amendment of our manuscript, we hope that this reviewer will appreciate the substantial progress that our study adds to the field.

Reviewer #3 (Remarks to the Author):

The authors describe a novel patterning methodology to utilize bulk cell density (BCD) as a key parameter to influence stem cell differentiation. The authors go on to conclude that BCD is an underappreciated (and underreported) factor in many publications on hPSC differentiation, and provide recommendations and methodologies for both reporting BCD within the culture format and also minimizing the effect of BCD by using specific culture format (monolayer expansion/differentiation platform). The authors provide a compelling story and a trove of data including flow cytometry, imaging, western blotting, protein arrays, mRNA microarray, and mass spectrometry to support their claims. Overall the study seems well designed, but given the expansive amount of data, there are experimental details lacking in the description of the study in some areas.

Specifically, I was asked to review the MS data collection methodologies for completeness and transparency. The description of

the measurement of CHIR99021 by mass spectrometry appears to have sufficient completeness and transparency. The below is a more complete review of the proteomics section. Much of the expanded details will likely need to be incorporated into supplemental information.

Insufficient details are provided in the Methods for the mass spectrometry and MS data analysis section for the cell culture supernatant. Cell culture supernatants are notoriously complex samples with wide dynamic range, therefore a gel image characteristic of the samples analyzed prior to band isolation would be a helpful reference. There are no details provided on the LC-MS/MS data collection parameters, such as chromatography setup (column, mobile phase, gradient, flow rate, temperature) or mass spectrometer settings (data acquisition resolution, speed, AGC settings, etc). Much of this could be clarified by providing the raw data available for download at one of several MS data repositories, such as Massive (<https://massive.ucsd.edu/ProteoSAFe/static/massive.jsp>). Additionally, the .fasta protein sequence database the authors used (ipi.HUMAN.v3.68) for database protein searching was last updated sometime prior to 2010, and is no longer available for download publicly.

This will make it difficult for anyone to reproduce the author's results even if they did have the raw data, and it is curious why the authors would choose to use a database and genome/proteome annotation that is more than six years old. By contrast, using the curated uniprot (www.uniprot.org) or NCBI curated database (<https://www.ncbi.nlm.nih.gov/protein>) would allow much more transparency. It is not clear whether the investigators searched against a database of common contaminants which could reduce false positive discoveries. Regardless of the database used, the authors should be required to provide a list of peptide (along with protein) identifications, including the monoisotopic m/z, retention time, any modifications on the peptide, and the database identification score. This is a minimum in order to assess results quality.

Response: We highly appreciate the comments of reviewer #3 and would like to apologize that we for reasons of space omitted essential information about mass spectrometric details.

As requested, we now provide more details about the chromatography setup and the mass spectrometer settings in the materials and methods section 'LC-MS/MS and automated MS data analysis' on page 18 of our manuscript as follows:

'Secreted proteins were precipitated from cell culture supernatants applying a carrier-assisted trichloroacetic acid precipitation 52 and subsequently separated by SDS-PAGE. Each gel lane was cut into small pieces that were subjected to tryptic digestion according to standard procedures 53. As described before 46 peptides were extracted, separated and analysed by reversed-phase chromatography using a nano-flow ultra-high-pressure liquid chromatography system (RSLC, Thermo Fisher Scientific) coupled online to an LTQ-Orbitrap Velos mass spectrometer (Thermo Fisher Scientific). For RSLC, peptides were trapped on a trapping column (3 μm C18 particle, 2 cm length, 75 μm inner diameter, Acclaim PepMap, Thermo Fisher Scientific) and separated using a reversed-phase separating column (2 μm C18 particle, 50 cm length, 75 μm inner diameter Acclaim PepMap, Thermo Fisher Scientific). Peptide elution was performed with a multi-step binary gradient: linear gradient of buffer B (80% ACN, 0.1% formic acid) in buffer A (0.1% formic acid) from 4% to 25% in 115 min, 25% to 50% in 25 min, 50% to 90% in 5 min, 90% B for 10 min and switched to 4% B within 30 min for reconditioning of the column at a flow rate of 250 ml/min and a column temperature of 45°C. For MS, overview scans were acquired at a resolution of 60,000 ($m/z = 400$) in the mass range of m/z 300 – 1600 and the 10 most intensive 2- and 3-fold charged ions were fragmented by collision-induced fragmentation at an activation time of 10 ms and an activation Q of 0.250 in the LTQ. Fragment ion spectra were recorded in the LTQ at a normal scan rate and stored as centroid m/z pairs. Raw data were processed with the MaxQuant proteomics software suit 54 version 1.5.0.22. Peak lists were searched against the human International uniprot protein sequence database (uniprot-homo+sapiens) downloaded on 09/07/2015 at www.uniprot.org at a false discovery rate of 1%. Proteins marked as "only identified by site", "reverse" or "contaminant" were removed from MaxQuant output files. Quantitative comparison between different samples was performed based on label-free quantification (LFQ) intensities calculated by MaxQuant.'

We added a figure in the supplemental material showing the gel of coomassie-stained proteins precipitated from culture supernatants, which was applied for excision of protein bands and subsequent mass spectrometric analysis (Supplemental Fig. 7).

We repeated the database search applying the curated uniprot database and further state in the material and methods section that we removed common contaminants from our protein lists (page 19, line 11-12). Figures 5b and c were updated, respectively.

As recommended we uploaded the LTQ Orbitrap .raw data files and .mzXML files, which were generated from these raw data as stated in the manuscript on page 19 as follows: 'The raw data were deposited to the ProteomeXchange Consortium via the PRIDE partner repository with the dataset identifier PXD004874. The processed data are listed in Supplementary Table 1.'
Reviewer login: Username: reviewer04674@ebi.ac.uk Password: K69yvXoc.

In addition, a table of protein identifications including the requested information about peptide identifications, monoisotopic m/z , retention time, modifications, score and further parameters was added as Supplementary Table 1.

References

- Alev, C., Wu, Y., Kasukawa, T., Jakt, L.M., Ueda, H.R., and Sheng, G. (2010). Transcriptomic landscape of the primitive streak. *Development* *137*, 2863-2874.
- Bertocchini, F., and Stern, C.D. (2002). The hypoblast of the chick embryo positions the primitive streak by antagonizing nodal signaling. *Developmental cell* *3*, 735-744.
- Blakeley, P., Fogarty, N.M., Del Valle, I., Wamaitha, S.E., Hu, T.X., Elder, K., Snell, P., Christie, L., Robson, P., and Niakan, K.K. (2015). Defining the three cell lineages of the human blastocyst by single-cell RNA-seq. *Development* *142*, 3613.
- Caneparo, L., Huang, Y.L., Staudt, N., Tada, M., Ahrendt, R., Kazanskaya, O., Niehrs, C., and Houart, C. (2007). Dickkopf-1 regulates gastrulation movements by coordinated modulation of Wnt/beta catenin and Wnt/PCP activities, through interaction with the Dally-like homolog Knypek. *Genes Dev* *21*, 465-480.
- Chen, B., Dodge, M.E., Tang, W., Lu, J., Ma, Z., Fan, C.W., Wei, S., Hao, W., Kilgore, J., Williams, N.S., *et al.* (2009). Small molecule-mediated disruption of Wnt-dependent signaling in tissue regeneration and cancer. *Nature chemical biology* *5*, 100-107.
- Funa, N.S., Schachter, K.A., Lerdrup, M., Ekberg, J., Hess, K., Dietrich, N., Honore, C., Hansen, K., and Semb, H. (2015). beta-Catenin Regulates Primitive Streak Induction through Collaborative Interactions with SMAD2/SMAD3 and OCT4. *Cell stem cell* *16*, 639-652.
- Hoepfner, J., Kleinsorge, M., Papp, O., Ackermann, M., Alfken, S., Rinas, U., Solodenko, W., Kirschning, A., Sgodda, M., and Cantz, T. (2016). Biphasic modulation of Wnt signaling supports efficient foregut endoderm formation from human pluripotent stem cells. *Cell biology international* *40*, 534-548.
- Hough, S.R., Thornton, M., Mason, E., Mar, J.C., Wells, C.A., and Pera, M.F. (2014). Single-cell gene expression profiles define self-renewing, pluripotent, and lineage primed states of human pluripotent stem cells. *Stem cell reports* *2*, 881-895.
- Itasaki, N., and Hoppler, S. (2010). Crosstalk between Wnt and bone morphogenic protein signaling: a turbulent relationship. *Developmental dynamics : an official publication of the American Association of Anatomists* *239*, 16-33.
- Iwashita, H., Shiraki, N., Sakano, D., Ikegami, T., Shiga, M., Kume, K., and Kume, S. (2013). Secreted cerberus1 as a marker for quantification of definitive endoderm differentiation of the pluripotent stem cells. *PLoS one* *8*, e64291.
- Joo, J.Y., Choi, H.W., Kim, M.J., Zaehres, H., Tapia, N., Stehling, M., Jung, K.S., Do, J.T., and Scholer, H.R. (2014). Establishment of a primed pluripotent epiblast stem cell in FGF4-based conditions. *Scientific reports* *4*, 7477.
- Kim, D.K., Cha, Y., Ahn, H.J., Kim, G., and Park, K.S. (2014). Lefty1 and lefty2 control the balance between self-renewal and pluripotent differentiation of mouse embryonic stem cells. *Stem cells and development* *23*, 457-466.
- Mendjan, S., Mascetti, V.L., Ortmann, D., Ortiz, M., Karjosukarso, D.W., Ng, Y., Moreau, T., and Pedersen, R.A. (2014). NANOG and CDX2 pattern distinct subtypes of human mesoderm during exit from pluripotency. *Cell stem cell* *15*, 310-325.
- Morrison, G.M., Oikonomopoulou, I., Migueles, R.P., Soneji, S., Livigni, A., Enver, T., and Brickman, J.M. (2008). Anterior definitive endoderm from ESCs reveals a role for FGF signaling. *Cell stem cell* *3*, 402-415.
- Niehrs, C. (2006). Function and biological roles of the Dickkopf family of Wnt modulators. *Oncogene* *25*, 7469-7481.
- O'Leary, T., Heindryckx, B., Lierman, S., van Bruggen, D., Goeman, J.J., Vandewoestyne, M., Deforce, D., de Sousa Lopes, S.M., and De Sutter, P. (2012). Tracking the progression of the human inner cell mass during embryonic stem cell derivation. *Nature biotechnology* *30*, 278-282.
- Pelletier, J.C., Lundquist, J.T.t., Gilbert, A.M., Alon, N., Bex, F.J., Bhat, B.M., Bursavich, M.G., Coleburn, V.E., Felix, L.A., Green, D.M., *et al.* (2009). (1-(4-(Naphthalen-2-yl)pyrimidin-2-yl)piperidin-4-yl)methanamine: a wingless beta-catenin agonist that increases bone formation rate. *Journal of medicinal chemistry* *52*, 6962-6965.
- Peng, G., Suo, S., Chen, J., Chen, W., Liu, C., Yu, F., Wang, R., Chen, S., Sun, N., Cui, G., *et al.* (2016). Spatial Transcriptome for the Molecular Annotation of Lineage Fates and Cell Identity in Mid-gastrula Mouse Embryo. *Developmental cell* *36*, 681-697.
- Perea-Gomez, A., Vella, F.D., Shawlot, W., Oulad-Abdelghani, M., Chazaud, C., Meno, C., Pfister, V., Chen, L., Robertson, E., Hamada, H., *et al.* (2002). Nodal antagonists in the anterior visceral endoderm prevent the formation of multiple primitive streaks. *Developmental cell* *3*, 745-756.
- Takehara, T., Teramura, T., Onodera, Y., Hamanishi, C., and Fukuda, K. (2012). Reduced oxygen concentration enhances conversion of embryonic stem cells to epiblast stem cells. *Stem cells and development* *21*, 1239-1249.
- Yu, P.B., Hong, C.C., Sachidanandan, C., Babbitt, J.L., Deng, D.Y., Hoynig, S.A., Lin, H.Y., Bloch, K.D., and Peterson, R.T. (2008). Dorsomorphin inhibits BMP signals required for embryogenesis and iron metabolism. *Nat Chem Biol* *4*, 33-41.

Reviewer #1 (Remarks to the Author)

The authors have made significant efforts in revising their work, also by carrying out additional experiments. Although I cannot agree to all responses to my previous points, I do acknowledge the fact that the criticism was appropriately taken into consideration. The abstract would still require some improvement.

Reviewer #3 (Remarks to the Author)

The authors have met the requirements for the requested revisions, and in my review the manuscript is now acceptable for publication.